# ADAPTIVE INERTIA: DISENTANGLING THE EFFECTS OF ADAPTIVE LEARNING RATE AND MOMENTUM

## ABSTRACT

Adaptive Momentum Estimation (Adam), which combines Adaptive Learning Rate and Momentum, would be the most popular stochastic optimizer for accelerating the training of deep neural networks. However, empirically Adam often generalizes worse than Stochastic Gradient Descent (SGD). We unveil the mystery of this behavior in the diffusion theoretical framework. Specifically, we disentangle the effects of Adaptive Learning Rate and Momentum of the Adam dynamics on saddle-point escaping and flat minima selection. We prove that Adaptive Learning Rate can escape saddle points efficiently, but cannot select flat minima as SGD does. In contrast, Momentum provides a drift effect to help the training process pass through saddle points, and almost does not affect flat minima selection. This partly explains why SGD (with Momentum) generalizes better, while Adam generalizes worse but converges faster. Furthermore, motivated by the analysis, we design a novel adaptive optimization framework named *Adaptive Inertia*, which uses parameter-wise adaptive inertia to accelerate the training and provably favors flat minima as well as SGD. Our extensive experiments demonstrate that the proposed adaptive inertia method can generalize significantly better than SGD and conventional adaptive gradient methods.

## 1 INTRODUCTION

Adam (Kingma and Ba, 2015), which combines Adaptive Learning Rate and Momentum, would be the most popular optimizer for accelerating the training of deep networks. However, Adam often generalizes worse and finds sharper minima than SGD (Wilson et al., 2017) for popular convolutional neural networks, where the flat minima have been argued to be closely related with good generalization (Hochreiter and Schmidhuber, 1995; 1997a; Hardt et al., 2016; Zhang et al., 2017).

Meanwhile, the diffusion theory has been used as a tool to study how SGD selects minima (Jastrzkebski et al., 2017; Li et al., 2017; Wu et al., 2018; Xu et al., 2018; Hu et al., 2019; Nguyen et al., 2019; Zhu et al., 2019; Xie et al., 2021b; Li et al., 2021). This line of research also suggests that injecting or enhancing SGD-like sharpness-dependent gradient noise may effectively help find flatter minima (An, 1996; Neelakantan et al., 2015; Zhou et al., 2019; Xie et al., 2021a;c; HaoChen et al., 2021). Especially, Zhou et al. (2020) argued the better generalization performance of SGD over Adam by showing that SGD enjoys smaller escaping time than Adam from a basin of the same local minimum. However, this argument does not reflect the whole picture of the dynamics of SGD and Adam. Empirically, it does not explain why Adam converges faster than SGD. Moreover, theoretically, all previous works have not touched the saddle-point escaping property of the dynamics, which is considered as an important challenge of efficiently training deep networks (Dauphin et al., 2014; Staib et al., 2019; Jin et al., 2017; Reddi et al., 2018).

Our work mainly has two contributions. **1)** We disentangle the effects of Adaptive Learning Rate and Momentum in Adam learning dynamics and characterize their behaviors in terms of saddle-point escaping and flat minima selection, which explains why Adam usually converges fast but does not generalize well. Particularly, we prove that, Adaptive Learning Rate is good at escaping saddle points but not good at selecting flat minima, while Momentum helps escape saddle point and matters little to escaping sharp minima[1]. **2)** We propose a novel Adaptive Inertia (Adai) optimization framework

---

[1]When we talk about saddle points, we mean strict saddle points in this paper.

Table 1: Test performance comparison of optimizers. We report the mean and the standard deviations (as the subscripts) of the optimal test errors computed over three runs. Our methods, including Adai and AdaiW, consistently outperforms all other popular optimizers.

| DATASET | MODEL | ADAIW | ADAI | SGD M | ADAM | AMSGRAD | ADAMW | ADABOUND | PADAM | YOGI | RADAM |
|---|---|---|---|---|---|---|---|---|---|---|---|
| CIFAR-10 | RESNET18 | $\mathbf{4.59}_{0.16}$ | $4.74_{0.14}$ | $5.01_{0.03}$ | $6.53_{0.03}$ | $6.16_{0.18}$ | $5.08_{0.07}$ | $5.65_{0.08}$ | $5.12_{0.04}$ | $5.87_{0.12}$ | $6.01_{0.10}$ |
|  | VGG16 | $\mathbf{5.81}_{0.07}$ | $6.00_{0.09}$ | $6.42_{0.02}$ | $7.31_{0.25}$ | $7.14_{0.14}$ | $6.48_{0.13}$ | $6.76_{0.12}$ | $6.15_{0.06}$ | $6.90_{0.22}$ | $6.56_{0.04}$ |
| CIFAR-100 | RESNET34 | $21.05_{0.10}$ | $\mathbf{20.79}_{0.22}$ | $21.52_{0.37}$ | $27.16_{0.55}$ | $25.53_{0.19}$ | $22.99_{0.40}$ | $22.87_{0.13}$ | $22.72_{0.10}$ | $23.57_{0.12}$ | $24.41_{0.40}$ |
|  | DENSENET121 | $\mathbf{19.44}_{0.21}$ | $19.59_{0.38}$ | $19.81_{0.33}$ | $25.11_{0.15}$ | $24.43_{0.09}$ | $21.55_{0.14}$ | $22.69_{0.15}$ | $21.10_{0.23}$ | $22.15_{0.36}$ | $22.27_{0.22}$ |
|  | GOOGLENET | $\mathbf{20.50}_{0.25}$ | $20.55_{0.32}$ | $21.21_{0.29}$ | $26.12_{0.33}$ | $25.53_{0.17}$ | $21.29_{0.17}$ | $23.18_{0.31}$ | $21.82_{0.17}$ | $24.24_{0.16}$ | $22.23_{0.15}$ |

which is conceptually orthogonal to the existing adaptive gradient framework. Adai does not parameter-wisely adjust learning rates, but parameter-wisely adjusts the momentum hyperparameter, called *inertia*. We show that Adai can provably escape saddle points fast while learning flat minima well. From the empirical results (see Table 1), Adai significantly outperforms SGD and popular Adam variants.

The paper is organized as follows. In Section 2, we first introduce the dynamics of SGD and analyze its behavior on saddle-point escaping, which serves as a basis to compare with the behavior of Adaptive Learning Rate and Momentum. In Section 3, we analyze the behavior of Momentum. In Section 4, we analyze the dynamics of Adam. In Section 5, we introduce the new optimizer Adai. In Section 6, we empirically compare the performance of Adai, Adam variants, and SGD. Section 7 concludes the paper with remarks.

## 2 SGD AND DIFFUSION

In this section, we introduce some preliminaries about the SGD diffusion and then present the saddle-point escaping property of SGD dynamics.

### 2.1 PREREQUISITES FOR SGD DIFFUSION

We first review the SGD diffusion theory for escaping minima proposed by Xie et al. (2021b). We denote the model parameters as $\theta$, the learning rate as $\eta$, the batch size as $B$, and the loss function over one minibatch and the whole training dataset as $\hat{L}(\theta)$ and $L(\theta)$, respectively. A typical optimization problem can be formulated as $\min_\theta L(\theta)$. We may write the stochastic differential equation/Langevin Equation that approximates SGD dynamics (Mandt et al., 2017; Li et al., 2019) as

$$d\theta = -\nabla L(\theta)dt + [\eta C(\theta)]^{\frac{1}{2}} dW_t, \tag{1}$$

where $dW_t \sim \mathcal{N}(0, Idt)$, $I$ is the identity matrix, and $C(\theta)$ is the gradient noise covariance matrix. The gradient noise is defined through the difference of the stochastic gradient over one minibatch and the true gradient over the whole training dataset, $\xi = \nabla \hat{L}(\theta) - \nabla L(\theta)$. It is well known that the Fokker-Planck Equation describes the probability density governed by Langevin Equation (Risken, 1996; Sato and Nakagawa, 2014). The Fokker-Planck Equation is

$$\frac{\partial P(\theta,t)}{\partial t} = \nabla \cdot [P(\theta,t)\nabla L(\theta)] + \nabla \cdot \nabla D(\theta)P(\theta,t), \tag{2}$$

where $\nabla \cdot$ is the divergence operator and $D(\theta) = \frac{\eta C(\theta)}{2}$ is the diffusion matrix (Xie et al., 2021b). We note that the dynamical time $t$ is equal to the product of the number of iterations $T$ and the learning rate $\eta$: $t = \eta T$.

As the gradient variance dominates the gradient expectation near critical points, we have

$$D(\theta) = \frac{\eta C(\theta)}{2} \approx \frac{\eta}{2B} \left[ \frac{1}{N} \sum_{j=1}^{N} \nabla L_j(\theta) \nabla L_j(\theta)^\top \right] = \frac{\eta}{2B} \text{FIM}(\theta) \approx \frac{\eta}{2B} [H(\theta)]^+ \tag{3}$$

near a critical point $c$, where $L_j(\theta)$ is the loss function of the $j$-th training sample, $H(\theta)$ is the Hessian of the loss function at $\theta$, and $\text{FIM}(\theta)$ is the observed Fisher Information matrix, referring to Chapter 8 of Pawitan (2001) and Zhu et al. (2019). We further verified that Equation (3) approximately

holds even not around critical points in Figure 6 of Appendix C. Equation (3) was also proposed by Jastrzkebski et al. (2017) and Zhu et al. (2019) and verified by Xie et al. (2021b) and Daneshmand et al. (2018). Please refer to Appendix C for the detailed analysis of the stochastic gradient noise.

If we have eigendecomposition $H = U \operatorname{diag}(H_1, \ldots, H_{n-1}, H_n)U^\top$, then we use $[\cdot]^+$ to denote the transformation that $[H]^+ = U \operatorname{diag}(|H_1|, \ldots, |H_{n-1}|, |H_n|)U^\top$. The $i$-th column vector of $U$ is the eigenvector of $H$ corresponding to the eigenvalue $H_i$.

In the following analysis, we use the second-order Taylor approximation near critical points. This assumption is common and mild, when we focus on the behaviors near critical points (Mandt et al., 2017; Zhang et al., 2019a; Xie et al., 2021b). Note that, by Equation (3) and Assumption 1, the diffusion matrix $D$ is independent of $\theta$ near critical points.

**Assumption 1.** *The loss function around the critical point $c$ can be approximately written as*

$$L(\theta) = L(c) + \frac{1}{2}(\theta - c)^\top H(c)(\theta - c).$$

## 2.2 SGD Diffusion near Saddle Points

In this subsection, we establish the saddle-point escaping property of SGD diffusion as Theorem 1.

**Theorem 1** (SGD Escapes Saddle Points). *Suppose $c$ is a critical point, Assumption 1 and Equation (3) hold, the dynamics is governed by Equation* (1)*, and the initial parameter is at the saddle point $\theta = c$. Then the probability density function of $\theta$ after time $t$ is given by the Gaussian distribution $\theta \sim \mathcal{N}(c, \Sigma(t))$, where $\Sigma(t) = U \operatorname{diag}(\sigma_1^2, \ldots, \sigma_{n-1}^2, \sigma_n^2)U^\top$ and*

$$\sigma_i^2(t) = \frac{D_i}{H_i}[1 - \exp(-2H_it)],$$

*where $D_i$ is the $i$-th eigenvalue of the diffusion matrix $D$ and $H_i$ is the $i$-th eigenvalue of the Hessian matrix $H$ at $c$. The column vectors of $U$ are exactly the eigenvectors of $H$. The dynamical time $t = \eta T$. In terms of SGD notations and $|H_i|\eta T \ll 1$ near saddle points, we have*

$$\sigma_i^2(T) = \frac{|H_i|\eta^2T}{B} + \mathcal{O}(B^{-1}H_i^2\eta^3T^2).$$

The proof is relegated to Appendix A.1. We note that 1) if $H_i > 0$, the distribution of $\theta$ along the direction $i$ converges to a Gaussian distribution with constant variance, $\mathcal{N}\left(c_i, \frac{\eta}{2B}\right)$; and 2) if $H_i < 0$, the distribution is a Gaussian distribution with the variance exponentially increasing with time $t$.

As the displacement $\Delta\theta_i$ from the saddle point $c$ can be modeled as a center-fixed Gaussian distribution, the mean squared displacement is equivalent to the variance, namely $\langle\Delta\theta_i^2(t)\rangle = \sigma_i^2(t)$. The result means that SGD escapes saddle points very slowly ($\langle\Delta\theta_i^2\rangle = \mathcal{O}(|H_i|)$) if $H_i$ is close to zero. Note that, in the diffusion analysis, the direction $i$ denotes the direction of an eigenvector instead of a coordinate's direction. While SGD updates model parameters along the coordinates, we do not need to treat the coordinates' directions specially in the continuous-time analysis.

## 3 Analysis of Momentum Dynamics

In this section, we analyze Momentum in terms of saddle-point escaping and flat minima selection.

**The continuous-time Momentum dynamics.** The Heavy Ball Method (Zavriev and Kostyuk, 1993) can be written as

$$\begin{cases} m_t = \beta_1 m_{t-1} + \beta_3 g_t, \\ \theta_{t+1} = \theta_t - \eta m_t, \end{cases} \tag{4}$$

where $\beta_1$ and $\beta_3$ are the hyperparameters. We note there are two popular choices, which are, respectively, $\beta_3 = 1$ corresponding to SGD-style Momentum and $\beta_3 = 1 - \beta_1$ corresponding to Adam-style Momentum, namely the exponentially moving average.

We can write the motion equation in physics with the mass $M$ and the damping coefficient $\gamma$ as

$$\begin{cases} r_t = (1 - \gamma dt)r_{t-1} + \frac{F}{M}dt \\ \theta_{t+1} = \theta_t + r_t dt, \end{cases} \tag{5}$$

where $r_t = -m_t$, $F = g_t$, $dt = \eta$, $1 - \gamma dt = \beta_1$, and $\frac{dt}{M} = \beta_3$. Thus, we obtain the differential form of the motion equation as

$$M\ddot{\theta} = -\gamma M\dot{\theta} + F, \tag{6}$$

where $\ddot{\theta} = \frac{d^2\theta}{dt^2}$ and $\dot{\theta} = \frac{d\theta}{dt}$. The left-hand side as the inertia force is equal to the mass $M$ times the acceleration $\ddot{\theta}$, the first term in the right-hand side as the damping force is equal to the damping coefficient $\gamma$ times the physical momentum $M\dot{\theta}$, and the second term in the right-hand side is equal to the external force $F$ in physics. We can easily obtain the mass $M = \frac{\eta}{\beta_3}$ and the damping coefficient $\gamma = \frac{1-\beta_1}{\eta}$ by comparing (5) and (6).

As $F$ corresponds to the stochastic gradient term, we obtain

$$Md\dot{\theta} = -\gamma Md\theta - \frac{\partial L(\theta)}{\partial \theta}dt + [2D]^{\frac{1}{2}}dW_t. \tag{7}$$

Its Fokker-Planck Equation in the phase space (the $\theta$-$\dot{\theta}$ space) is well known as

$$\frac{\partial P(\theta, r, t)}{\partial t} = -\nabla_\theta \cdot [rP(\theta, r, t)] + \nabla_r \cdot \left[\gamma r + M^{-1}\nabla_\theta L(\theta)\right] P(\theta, r, t)$$
$$+ \nabla_r \cdot M^{-2}D \cdot \nabla_r P(\theta, r, t) \tag{8}$$

where $r = \dot{\theta}$. Equation (8) is not specialized for learning dynamics but a popular result in Langevin Dynamics literatures (Risken, 1996; Risken and Eberly, 1985; Zhou, 2010; Kalinay and Percus, 2012). Equation (57) of Radpay (2020) gives an exactly same form for describing finite-inertia Langevin Dynamics, Equation (7). We contribution is the first to apply it to deep learning dynamics.

**Momentum escapes saddle points.** We formulate how Momentum escapes saddle points as Theorem 2, whose proof is relegated to Appendix A.2.

**Theorem 2** (Momentum Escapes Saddle Points). *Suppose $c$ is a critical point, Assumption 1 and Equation* (3) *hold, the dynamics is governed by Momentum, and the initial parameter is at the saddle point $\theta = c$. Then the mean squared displacement near saddle points is*

$$\langle \Delta\theta_i^2(t) \rangle = \frac{D_i}{\gamma^3 M^2}[1 - \exp(-\gamma t)]^2 + \frac{D_i}{\gamma M H_i}[1 - \exp(-\frac{2H_i t}{\gamma M})], \tag{9}$$

*where $\Delta\theta(t) = \theta(t) - \theta(0)$ is the displacement of $\theta$, and $\langle \cdot \rangle$ denote the mean value. In the right-hand side, this first term is the momentum drift effect and the second term is the diffusion effect. As $|H_i|\eta T \ll 1$ near ill-conditioned saddle points, it can be written in terms of Momentum notations as*

$$\langle \Delta\theta_i^2 \rangle = \frac{|H_i|\beta_3^2\eta^2}{2(1-\beta_1)^3 B}\left[1 - \exp\left(-(1-\beta_1)T\right)\right]^2 + \frac{|H_i|\beta_3^2\eta^2 T}{B(1-\beta_1)^2} + \mathcal{O}(B^{-1}H_i^2\eta^3 T^2).$$

By comparing Theorems 1 and 2, we notice that, SGD escapes saddle points only due to the diffusion effect (similar to the second term in Equation (9)), but Momentum provides an additional momentum drift effect (the first term in Equation (9)) for passing through saddle points (Wang et al., 2019). This momentum drift effect has not been mathematically revealed before, to the best of our knowledge.

**Momentum escapes minima.** We first introduce two classical assumptions.

**Assumption 2** (Quasi-Equilibrium Approximation). *The system is in quasi-equilibrium near minima.*

**Assumption 3** (Low Temperature Approximation). *The system is under low temperature (small gradient noise).*

Xie et al. (2021b) modeled the process of SGD escaping minima as a Kramers Escape Problem in deep learning. Recent machine learning papers (Jastrzkebski et al., 2017; Xie et al., 2021b; Zhou et al., 2020) also used Assumptions 2 and 3 as the background implicitly or explicitly. Quasi-Equilibrium Approximation and Low Temperature Approximation have been widely used in many fields' Kramers Escape Problems for state transition/minima selection, including statistical physics(Kramers, 1940; Hanggi, 1986), chemistry(Eyring, 1935; Hänggi et al., 1990), biology(Zhou, 2010), electrical engineering(Coffey and Kalmykov, 2012), and stochastic process(Van Kampen, 1992; Berglund, 2013).

Assumptions 2 and 3 mean that the diffusion theory is good at modeling "slow" escape processes that cost more iterations. As this class of "slow" escape processes takes main computational time compared with "fast" escape processes, this class of "slow" escape process is more interesting for training of deep neural networks. Empirically, Xie et al. (2021b) reported that the escape processes in the wide range of iterations (50 to 100,000 iterations) can be modeled as a Kramers Escape Problem very well. Our empirical results in Section 6 support this point again. Thus, Assumptions 2 and 3 are reasonable in practice (see more discussions in Appendix B).

We next formulate how Momentum escapes minima as Theorem 3, which is based on Theorem 3.2 in Xie et al. (2021b) and the effective diffusion correction for the phase-space Fokker-Planck Equation (8) in Kalinay and Percus (2012). We analyze the mean escape time $\tau$ (Van Kampen, 1992; Xie et al., 2021b) required for the escaping process from a loss valley. Suppose that the saddle point $b$ is the exit of Loss Valley $a$, and $\Delta L = L(b) - L(a)$ is the barrier height(See more details in Appendix E.).

**Theorem 3** (Momentum Escapes Minima). *Suppose Assumptions 1, 2, and 3 hold, and the dynamics is governed by Momentum. Then the mean escape time from Loss Valley $a$ to the outside of Loss Valley $a$ is given by*

$$\tau = \pi \left[ \sqrt{1 + \frac{4|H_{be}|}{\gamma^2 M}} + 1 \right] \frac{1}{|H_{be}|} \cdot \exp\left[ \frac{2\gamma M B \Delta L}{\eta} \left( \frac{s}{H_{ae}} + \frac{(1-s)}{|H_{be}|} \right) \right],$$

*where the subscript $e$ indicates the escape direction, $s \in (0,1)$ is a path-dependent parameter, and $H_{ae}$ and $H_{be}$ are the eigenvalues of the Hessians of the loss function at the minimum $a$ and the saddle point $b$ along the escape direction $e$. When $\frac{4|H_{be}|}{\gamma^2 M} \ll 1$, it reduces to SGD. In terms of Momentum notations, we have $\log(\tau) = \mathcal{O}\left( \frac{2(1-\beta_1)B\Delta L}{\beta_3 \eta H_{ae}} \right)$.*

Note that the escape direction is aligned with some eigenvector in the diffusion theory (Xie et al., 2021b). We leave the proof in Appendix A.3. We note that $\log(\tau) = \mathcal{O}\left( \frac{2B\Delta L}{\eta H_{ae}} \right)$ has been obtained for SGD (Xie et al., 2021b). We see that Momentum does not affect flat minima selection in terms of the mean escape time, if we properly choose the learning rate, i.e., $\eta_{\text{Momentum}} = \frac{1-\beta_1}{\beta_3} \eta_{\text{SGD}}$.

## 4 ANALYSIS OF ADAM DYNAMICS

| **Algorithm 1:** Adam |
| --- |
| $g_t = \nabla \hat{L}(\theta_t)$; 
 $m_t = \beta_1 m_{t-1} + (1-\beta_1)g_t$; 
 $v_t = \beta_2 v_{t-1} + (1-\beta_2)g_t^2$; 
 $\hat{m}_t = \frac{m_t}{1-\beta_1^t}$; 
 $\hat{v}_t = \frac{v_t}{1-\beta_2^t}$; 
 $\theta_{t+1} = \theta_t - \frac{\eta}{\sqrt{\hat{v}_t}+\epsilon}\hat{m}_t$; |

| **Algorithm 2:** Adai |
| --- |
| $g_t = \nabla \hat{L}(\theta_t)$; 
 $v_t = \beta_2 v_{t-1} + (1-\beta_2)g_t^2$; 
 $\hat{v}_t = \frac{v_t}{1-\beta_2^t}$; 
 $\bar{v}_t = \text{mean}(\hat{v}_t)$; 
 $\beta_{1,t} = (1 - \frac{\beta_0}{\bar{v}_t}\hat{v}_t).\text{Clip}(0, 1-\epsilon)$; 
 $m_t = \beta_{1,t} m_{t-1} + (1-\beta_{1,t})g_t$; 
 $\hat{m}_t = \frac{m_t}{1-\prod_{z=1}^{t}\beta_{1,z}}$; 
 $\theta_{t+1} = \theta_t - \eta\hat{m}_t$; |

In this section, we analyze the effects of Adaptive Learning Rate in terms of saddle-point escaping and flat minima selection.

**A motivation behind Adaptive Learning Rate.** The previous theoretical results naturally point us a good way to help escape saddle points: adaptively adjust the learning rates for different parameter as $\eta_i \propto |H_i|^{-\frac{1}{2}}$. However, estimating the Hessian is expensive in practice. In Adam (Algorithm 1), the diagonal $v$ can be regarded as a diagonal approximation of the full matrix $\mathbb{E}[g_t g_t^\top]$ (Staib et al., 2019). Following Staib et al. (2019), our analysis considers the "idealized" Adam where $v = \mathbb{E}[g_t g_t^\top]$ is a full matrix. Note that $\mathbb{E}[g_t g_t^\top] = C(\theta) = \frac{H}{B}$ approximately holds near critical points.

**The continuous-time Adam dynamics.** The continuous-time dynamics of Adam can also be written as Equation (6), except the mass $M = \frac{\hat{\eta}}{1-\beta_1}$ and the damping coefficient $\gamma = \frac{1-\beta_1}{\hat{\eta}}$. We emphasize that the learning rate is not a real number but the diagonal approximation of the ideal learning rate

matrix $\hat{\eta} = \eta C^{-\frac{1}{2}}$ in practice. We apply the adaptive time continuation $d\hat{t}_i = \hat{\eta}_i$ in the $i$-th dimension, where $\hat{t}_i$ of $T$ iterations is defined as the sum of $\hat{\eta}_i$ of each iteration. Thus, the Fokker-Planck Equation for Adam can still be written as Equation (8). We leave more details in Appendix H.

**Adam escapes saddle points.** Similarly to Theorem 2, we formulate how Adam escapes saddle point as Proposition 1, which can be obtained from Theorem 2 and $\hat{\eta} = \eta C^{-\frac{1}{2}}$.

**Proposition 1** (Adam Escapes Saddle Points). *Suppose $c$ is a critical point, Assumption 1 and Equation* (3) *hold, the dynamics is governed by Adam, and the initial parameter is at the saddle point $\theta = c$. Then the mean squared displacement is written as*

$$\langle \Delta\theta_i^2(\hat{t}_i) \rangle = \frac{D_i}{\gamma^2 M}[1 - \exp(-\gamma\hat{t}_i)]^2 + \frac{D_i}{\gamma M H_i}[1 - \exp(-\frac{2H_i\hat{t}_i}{\gamma M})].$$

*Under $|H_i|\eta T \ll 1$, it can be written in terms of Adam notations as*

$$\langle \Delta\theta_i^2 \rangle = \frac{\eta^2}{2(1-\beta_1)}\left[1 - \exp\left(-(1-\beta_1)T\right)\right]^2 + \eta^2 T + \mathcal{O}(\sqrt{B|H_i|}\eta^3 T^2).$$

From Proposition 1, we can see that Adam escapes saddle points fast, because both the momentum drift and the diffusion effect are approximately Hessian-independent and isotropic near saddle points, which is also supported by Staib et al. (2019). Proposition 1 indicates that one advantage of Adam over RMSprop (Hinton et al., 2012) comes from the additional momentum drift effect on saddle-point escaping.

**Adam escapes minima.** We next formulate how Adam escapes minima as Proposition 2, which shows that Adam cannot learn flat minima as well as SGD. We leave the proof in Appendix A.4.

**Proposition 2** (Adam Escapes Minima). *The mean escape time of Adam only exponentially depends on the square root of the eigenvalues of the Hessian at a minimum:*

$$\tau = \pi\left[\sqrt{1 + \frac{4\eta\sqrt{B|H_{be}|}}{1-\beta_1}} + 1\right]\frac{|\det(H_a^{-1}H_b)|^{\frac{1}{4}}}{|H_{be}|}\exp\left[\frac{2\sqrt{B}\Delta L}{\eta}\left(\frac{s}{\sqrt{H_{ae}}} + \frac{(1-s)}{\sqrt{|H_{be}|}}\right)\right],$$

*where $\tau$ is the mean escape time. Thus, we have $\log(\tau) = \mathcal{O}\left(\frac{2\sqrt{B}\Delta L}{\eta\sqrt{H_{ae}}}\right)$.*

In comparison, Adam has $\log(\tau) = \mathcal{O}(H_{ae}^{-\frac{1}{2}})$, while SGD and Momentum both have $\log(\tau) = \mathcal{O}(H_{ae}^{-1})$. We say that Adam is weaker *Hessian-dependent* than SGD in terms of sharp minima escaping, which means that if $H_{ae}$ increases, i.e., the escaping direction becomes steeper, the escaping time $\log(\tau)$ for SGD decreases more dramatically than $\log(\tau)$ for Adam. Note that, in the diffusion theory, the minima sharpness is reflected by $H_{ae}$, the eigenvalue of the Hessian corresponding to the escape direction along an eigenvector. The weaker Hessian-dependent diffusion term of Adam hurts the efficiency of selecting flat minima in comparison with SGD. In summary, Adam is better at saddle-point escaping but worse at flat minima selection than SGD.

**Adam variants.** Many variants of Adam have been proposed to improve performance, such as AdamW (Loshchilov and Hutter, 2018), AdaBound (Luo et al., 2019), Padam (Chen and Gu, 2018), RAdam (Liu et al., 2019), AMSGrad (Reddi et al., 2019), Yogi (Zaheer et al., 2018), and others (Shi et al., 2021; Défossez et al., 2020; Zou et al., 2019). Many of them introduced extra hyperparameters that require tuning effort. Most variants often generalize better than Adam, while they may still not generalize as well as fine-tuned SGD (Zhang et al., 2019b), which we also discuss in Appendix F. Moreover, they do not have theoretical understanding of minima selection. Loosely speaking, our analysis suggests that Adam variants may also be poor at selecting flat minima due to Adaptive Learning Rate.

## 5 ADAPTIVE INERTIA

In this section, we propose a novel adaptive inertia optimization framework.

**Adaptive Inertia Optimization (Adai).** The basic idea of Adai (Algorithm 2) comes from Theorems 2 and 3, from which we can see that parameter-wise adaptive inertia can achieve the approximately

Hessian-independent momentum drift without damaging flat minima selection. The total momentum drift effect during passing through a saddle point in Theorem 2 is given by

$$\langle \Delta\theta_i \rangle^2 = \frac{|H_i|\eta^2}{2(1-\beta_1)B}. \tag{10}$$

We generalize the scalar $\beta_1$ in Momentum to a vector $\beta_1$ in Adai. The ideal adaptive updating is to keep $\beta_1 = 1 - \frac{\beta_0}{\bar{v}}v$, where the rescaling factor $\bar{v}$, namely the mean of all elements in the estimated $\hat{v}$, could be used to make the optimizer more robust in practice. The default values of $\beta_0$ and $\beta_2$ are 0.1 and 0.99, respectively. The hyperparameter $\epsilon$ is for avoiding extremely large inertia. The default setting $\epsilon = 0.001$ means the maximal inertia is 1000 times the minimal inertia, as $M_{\max} = \eta(1 - \beta_{1,\max})^{-1} = \eta\epsilon^{-1}$. Compared with Adam, Adai does not increase the number of hyperparameters. Note that an existing "adaptive momentum" method (Wang and Ye, 2020) is not parameter-wisely adaptive and essentially different from Adai.

The Fokker-Planck Equation associated with Adai dynamics in the phase space can be written as Equation (8), except that we replace the mass coefficient and the damping coefficient by the mass matrix and the damping matrix: $M = \eta(I - \text{diag}(\beta_1))^{-1}$ and $\gamma = \eta^{-1}(I - \text{diag}(\beta_1))$.

**Adai escapes saddle points.** Proposition 3 shows that Adai can escape saddle points efficiently due to an isotropic and approximately Hessian-independent momentum drift, while the diffusion effect is the same as Momentum. Proposition 3 is a direct result of Theorem 2.

**Proposition 3** (Adai Escapes Saddle Points). *Suppose $c$ is a critical point, Assumption 1 and Equation (3) hold, the dynamics is governed by Adai, and the initial parameter is at the saddle point $\theta = c$. Then the total momentum drift in the procedure of passing through the saddle point is given by*

$$\langle \Delta\theta_i \rangle^2 = \frac{\bar{v}\eta^2}{\beta_0} = \frac{\sum_{i=1}^n |H_i|\eta^2}{\beta_0 nB},$$

*where $\sum_{i=1}^n |H_i|$ is the trace norm of the Hessian at $c$.*

Intuitively, the momentum drift effect of Adai can be significantly larger than Adam by allowing larger inertia, which can be verified in experiments. We do not rigorously claim that Adai must converge faster than or as fast as Adam. Instead, we will empirically compare Adai with Adam on various datasets.

**Adai escapes minima.** We formulate how Adai escapes minima in Proposition 4, which shows Adai can learn flat minima better than Adam. We leave the proof in Appendix A.5.

**Proposition 4** (Adai Escapes Minima). *The mean escape time of Adai exponentially depends on the eigenvalues of the Hessian at a minimum:*

$$\tau = \pi \left[ \sqrt{1 + \frac{4\eta\sum_{i=1}^n |H_{bi}|}{\beta_0 n}} + 1 \right] \frac{1}{|H_{be}|} \exp\left[ \frac{2B\Delta L}{\eta}\left( \frac{s}{H_{ae}} + \frac{(1-s)}{|H_{be}|} \right) \right],$$

*where $\tau$ is the mean escape time. Thus, we have $\log(\tau) = \mathcal{O}\left( \frac{2B\Delta L}{\eta H_{ae}} \right)$.*

We can see that both SGD and Adai have $\log(\tau) = \mathcal{O}(H_{ae}^{-1})$, while Adam has $\log(\tau) = \mathcal{O}(H_{ae}^{-\frac{1}{2}})$. Thus, Adai favors flat minima as well as SGD and better than Adam.

**Convergence Analysis.** Theorem 4 proves that Adai has similar convergence guarantees to SGD Momentum(Yan et al., 2018; Ghadimi and Lan, 2013). The proof is relegated to Appendix A.6.

**Theorem 4** (Convergence of Adai). *Assume that $L(\theta)$ is an $\mathcal{L}$-smooth function[2], $L$ is lower bounded as $L(\theta) \geq L^\star$, $\mathbb{E}[\xi] = 0$, $\mathbb{E}[\|g(\theta,\xi) - \nabla L(\theta)\|^2] \leq \delta^2$, and $\|\nabla L(\theta)\| \leq G$ for any $\theta$, where $\xi$ represents the gradient noise of sub-sampling. Let Adai run for $t + 1$ iterations and $\beta_{1,\max} = 1 - \epsilon \in [0, 1)$ for any $t \geq 0$. If $\eta \leq \frac{C}{\sqrt{t+1}}$, we have*

$$\min_{k=0,\dots,t} \mathbb{E}[\|\nabla L(\theta_k)\|^2] \leq \frac{1}{\sqrt{t+1}}(C_1 + C_2 + C_3),$$

*where $C_1 = \frac{L(\theta_0) - L^\star}{(1-\beta_{1,\max})C}$, $C_2 = \frac{\beta_{1,\max}C}{2(1-\beta_{1,\max})^2}G^2$, and $C_3 = \frac{\mathcal{L}C}{2(1-\beta_{1,\max})^2}(G^2 + \delta^2)$.*

---

[2]It means that $\|\nabla L(\theta_a) - \nabla L(\theta_b)\| \leq \mathcal{L}\|\theta_a - \theta_b\|$ holds for any $\theta_a$ and $\theta_b$.

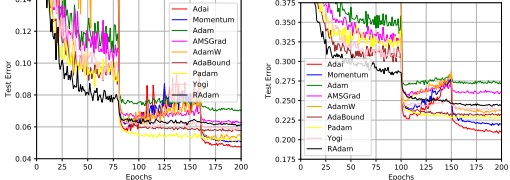

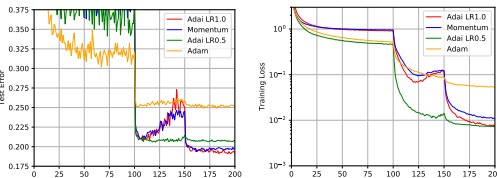

Figure 1: The learning curves on CIFAR-10 and CIFAR-100. Left: ResNet18 on CIFAR-10. Right: ResNet34 on CIFAR-100.

Figure 2: Adai shows better Generalization in the comparison with Momentum and Adam under similar convergence speed. DenseNet121 on CIFAR-100.

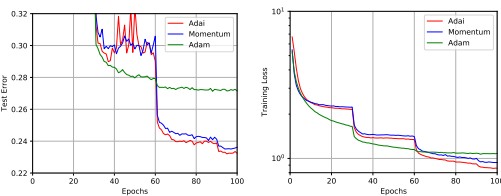

|      | Adai        | SGD M | Adam  |
|------|-------------|-------|-------|
| Top1 | **23.20**   | 23.51 | 27.13 |
| Top5 | **6.62**    | 6.82  | 9.18  |

Figure 3: ResNet50 on ImageNet. Left Subfigure: Top 1 Test Error. Right Subfigure: Training Loss. Table: Top-1 and top-5 test errors. Note that the popular SGD baseline performance of ResNet50 on ImageNet has the test errors as 23.85% in PyTorch and 24.9% in He et al. (2016), which are both weaker than our SGD baseline.

We note that Adai is the base optimizer in the adaptive inertia framework, while Adam is the base optimizer in the adaptive gradient framework. We can also combine Adai with stable/decoupled weight decay (AdaiS/AdaiW) (Loshchilov and Hutter, 2018; Xie et al., 2020) (see more discussions in Appendix G.) or other techniques from adaptive gradient methods.

## 6 EMPIRICAL ANALYSIS

In this section, we first conduct experiments to compare Adai, Adam, and SGD with Momentum in terms of convergence speed and generalization, and then empirically analyze flat minima selection.

**Datasets:** CIFAR-10, CIFAR-100(Krizhevsky and Hinton, 2009), ImageNet(Deng et al., 2009), and Penn TreeBank(Marcus et al., 1993). **Models:** ResNet18/ResNet34/ResNet50 (He et al., 2016), VGG16 (Simonyan and Zisserman, 2014), DenseNet121 (Huang et al., 2017), GoogLeNet (Szegedy et al., 2015), and Long Short-Term Memory (LSTM) (Hochreiter and Schmidhuber, 1997b). More details and results can be found in Appendix D and Appendix F.

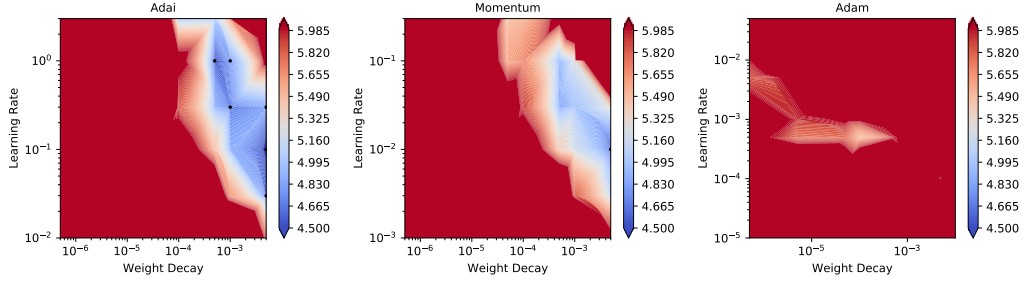

Figure 4: The test errors of ResNet18 on CIFAR-10 under various learning rates and weight decay. Adai has a much deeper and wider blue region near dark points ($\leq 4.83\%$) than SGD with Momentum and Adam.

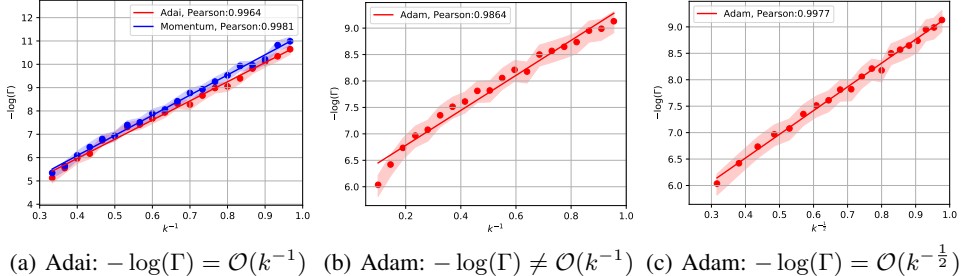

(a) Adai: $-\log(\Gamma) = \mathcal{O}(k^{-1})$  (b) Adam: $-\log(\Gamma) \neq \mathcal{O}(k^{-1})$  (c) Adam: $-\log(\Gamma) = \mathcal{O}(k^{-\frac{1}{2}})$

Figure 5: Flat Minima Selection: $Adai \approx Momentum \gg Adam$. The log-scale mean escape time $-\log(\Gamma)$ with the $95\%$ confidence interval is displayed. We empirically verify that $-\log(\Gamma) = \mathcal{O}(k^{-1})$ holds for Momentum and Adai but does not hold for Adam. Instead, we observe that $-\log(\Gamma) = \mathcal{O}(k^{-\frac{1}{2}})$ holds better for Adam. While Adai and Momentum favor flat minima similarly, Adai may escape loss valleys slightly faster than Momentum.

**Generalization and Convergence Speed.** We display the test performance of all models on CIFAR in Table 1. Table 1 and Figure 1 shows that Adai has excellent generalization compared with other popular optimizers on CIFAR-10/CIFAR-100. Figure 2 further demonstrates that Adai can consistently generalize better than SGD with Momentum and Adam, while maintaining similarly fast or faster convergence, respectively.

**Image Classification on ImageNet.** Figure 3 shows that Adai generalizes significantly better than SGD and Adam by training ResNet50 on ImageNet.

**Robustness to the hyperparameters.** Figure 4 demonstrates that Adai not only has better optimal test performance, but also has a wider test error basin than SGD with Momentum and Adam. It means that Adai is more robust to the choices of learning rates and weight decay.

**The Mean Escape Time Analysis.** We empirically study how the escape rate $\Gamma$, which equals to the inverse mean escape time, depends on the minima sharpness for different optimizers in Figure 5. We use Styblinski-Tang Function as the test function which has clear boundaries between loss valleys. Our method for adjusting the minima sharpness is to multiply a rescaling factor $\sqrt{k}$ to each parameter, and the Hessian will be proportionally rescaled by a factor $k$. If we let $L(\theta) = f(\theta) \rightarrow L(\theta) = f(\sqrt{k}\theta)$, then $H(\theta) = \nabla^2 f(\theta) \rightarrow H(\theta) = k\nabla^2 f(\theta)$. Thus, we can use $k$ to indicate the relative minima sharpness. We leave more details in Appendix E.

From Figure 5, it is easy to see that we have $\log(\tau) = \mathcal{O}(H_{ae}^{-1})$ in Adai and SGD, while we have $\log(\tau) = \mathcal{O}(H_{ae}^{-\frac{1}{2}})$ in Adam. Moreover, Adam is significantly less dependent on the minima sharpness than both Momentum and Adai.

**Supplementary Experiments. (1)** It is known that training of language models requires Adaptive Learning Rate (Zhang et al., 2019b). We will leave fusing Adaptive Learning Rate and Adaptive Inertia together as future work, which may be interesting for Transformer-based models (Vaswani et al., 2017). We leave the results of training LSTM on Penn TreeBank in Figure 12 of Appendix F. **(2)** In Appendix F, we used the expected minima sharpness in Neyshabur et al. (2017) and empirically verified again that Adai and Momentum can learn flatter minima than Adam.

# 7    CONCLUSION

To the best of our knowledge, we are the first to theoretically disentangle the effects of Adaptive Learning Rate and Momentum in terms of saddle-point escaping and flat minima selection. Under reasonable assumptions, our theory explains why Adam is good at escape saddle points but not good at selecting flat minima. We further propose a novel optimization framework, Adai, which can parameter-wisely adjust the momentum hyperparameter. Supported by good theoretical properties, Adai can accelerate training and favor flat minima well at the same time. Our empirical analysis demonstrates that Adai generalizes significantly better than popular Adam variants and SGD.

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

## A    PROOFS

### A.1    PROOF OF THEOREM 1

*Proof.* It is easy to validate that the probability density function

$$P(\theta, t) = \frac{1}{\sqrt{(2\pi)^n \det(\Sigma(t))}} \exp\left(-\frac{1}{2}(\theta - c)^\top \Sigma(t)(\theta - c)\right) \tag{11}$$

is the solution of the Fokker-Planck Equation (2). Without losing generality, we only validate one-dimensional solution, such as Dimension $i$.

The first term in Equation (2) can be written as

$$\frac{\partial P(\theta, t)}{\partial t} = -\frac{1}{2}\frac{1}{\sqrt{2\pi\sigma^2}}\frac{1}{\sigma^2}\exp\left(-\frac{\theta^2}{2\sigma^2}\right)\frac{\partial \sigma^2}{\partial t} + \frac{1}{\sqrt{2\pi\sigma^2}}\exp\left(-\frac{\theta^2}{2\sigma^2}\right)\frac{\theta^2}{2\sigma^4}\frac{\partial \sigma^2}{\partial t} \tag{12}$$

$$= \frac{1}{2}\left(\frac{\theta^2}{\sigma^4} - \frac{1}{\sigma^2}\right)P(\theta, t)\frac{\partial \sigma^2}{\partial t}. \tag{13}$$

The second term in Equation (2) can be written as

$$\nabla \cdot [P(\theta,t)\nabla L(\theta)] = P(\theta)H + H\theta \frac{1}{\sqrt{2\pi\sigma^2}}\exp\left(-\frac{\theta^2}{2\sigma^2}\right)\left(-\frac{\theta}{\sigma^2}\right) \tag{14}$$

$$= H\left(1 - \frac{\theta^2}{\sigma^2}\right)P(\theta,t). \tag{15}$$

The third term in Equation (2) can be written as

$$D\nabla^2 P(\theta,t) = -D\frac{\sigma^2 - \theta^2}{\sigma^5\sqrt{2\pi}}\exp\left(-\frac{\theta^2}{2\sigma^2}\right) \tag{16}$$

$$= D\left(\frac{\theta^2}{\sigma^4} - \frac{1}{\sigma^2}\right)P(\theta,t). \tag{17}$$

By $Term1 = Term2 + Term3$, we have

$$\frac{1}{2}(\theta^2 - \sigma^2)\frac{\partial\sigma^2}{\partial t} = H\sigma^2(\sigma^2 - \theta^2) + D(\theta^2 - \sigma^2) \tag{18}$$

$$\frac{\partial\sigma^2}{\partial t} = 2D - 2H\sigma^2. \tag{19}$$

The initial condition of $\sigma^2$ is given by $\sigma^2(0) = 0$. We can validate that $\sigma^2$ satisfies

$$\sigma_i^2(t) = \frac{D_i}{H_i}[1 - \exp(-2H_i t)]. \tag{20}$$

It is true along all eigenvectors' directions.

By $D = \frac{\eta}{2B}H$, we can get the results of SGD diffusion:

$$\sigma_i^2(T) = \text{sign}(H_i)\frac{\eta}{2B}[1 - \exp(-2H_i\eta T)] \tag{21}$$

$\square$

## A.2   PROOF OF THEOREM 2

*Proof.* Without losing the generality, we consider the one-dimensional case that aligns with an eigenvector of the Hessian. The stationary solution in equilibrium to the phase-space Fokker-Planck Equation near a critical point is given by a canonical ensemble

$$P_{eq}(\theta, r) = Z^{-1}\exp[-\beta(L(\theta) + \frac{1}{2}Mr^2)], \tag{22}$$

where $Z$ is a partition function and the inverse temperature $\beta = \gamma M D^{-1}$. This result is famous in a large number of physics literature (Van Kampen, 1992; Risken, 1996; Balakrishnan, 2008). Thus, we have the equilibrium distribution in velocity space and position space as

$$P_{eq}(r) = Z_r^{-1}\exp(-\frac{\beta M r^2}{2}) \tag{23}$$

and

$$P_{eq}(\theta) = Z_r^{-1}\exp(-\beta L(\theta)), \tag{24}$$

respectively, where $Z_r$ is the partition function for $P_{eq}(r)$ and $Z_\theta$ is the partition function for $P_{eq}(\theta)$.

It is known that the phase-space probability density solution $P(\theta, r, t)$ can be obtained from the position-space solution $P(\theta, t)$ and the velocity-space solution $P(r, t)$. Thus, we may compute the velocity-space solution (in Step I) and the position-space solution (in Step II), respectively.

Step I:

We further argue that the velocity-space solution $P(r, t)$ has reached its equilibrium Boltzmann distribution $P_{eq}(r)$ when the particle is passing saddle points. This is reasonable because the equilibrium solution in the velocity space is stable under Assumption 1. The velocity distribution $P(r, t)$ must be the solution of the free diffusion equation in the velocity space. We can approximately ignore the gradient expectation term $-\frac{\partial L(\theta)}{\partial \theta}$ in dynamical equations near critical points, as the gradient expectation is much smaller the gradient noise scale near critical points. Near critical points, the velocity $r$ obeys the equilibrium distribution

$$P(r, t) \approx P_{eq}(r) = (2\pi)^{-\frac{n}{2}} \det(\beta M)^{\frac{1}{2}} \exp(-\frac{\beta M r^2}{2}), \tag{25}$$

where $n = 1$ in one-dimensional case. This is a classical Boltzmann distribution. The expected velocity (also called "equilibrium velocity") along dimension $i$ can be given by $r_{eq,i}^2 = \frac{D_i}{\gamma M^2}$.

Step II:

As we discussed in Theorem 1, the position-space distribution in equilibrium is time-dependent near saddle points. Following the form of the solution to the position-space Fokker-Planck Equation (Seen in Appendix A.1), the ansatz solution of $P(\theta, t)$ is given by

$$\begin{cases} P(\theta, t) = \frac{1}{\sqrt{(2\pi)^n \det(\Sigma(t))}} \exp\left(-\frac{1}{2}(\theta - c(t))^\top \Sigma(t)(\theta - c(t))\right) \\ \Sigma(t) = U \operatorname{diag}(\sigma_1^2(t), \ldots, \sigma_{n-1}^2(t), \sigma_n^2(t)) U^\top \end{cases}$$

The time-dependent components in $P(\theta, t)$ are caused by two effects. The first effect is the momentum drift effect (due to equilibrium velocity), which decides $c(t)$, the center position of the probability density. As we studied in Theorem 1, the second effect is the diffusion effect (due to random noise), which decides the covariance of the probability density, $\Sigma(t)$.

The momentum drift effect is governed by the motion equation without random noise, while the diffusion effect is governed by the diffusion equation where gradient noise dominates gradient mean.

Step II (a): the momentum drift effect.

We can write the dynamics of the momentum drift effect as

$$M\ddot{c}(t) = -\gamma M \dot{c}(t). \tag{26}$$

We also have ignored the conservative force $-H[c(t) - c(0)]$ near saddle points, as $|H_i| \ll \gamma M$ exists for ill-condition Hessian eigenvalues. We focus on the behaviors near ill-conditioned saddle points, where Hessian eigenvalues are small along the escape directions.

The initial condition is given by $c(0) = c$, $\dot{c}(0) = r_{eq}$, and $\ddot{c}(0) = -\gamma r_{eq}$. Then we can obtain the solution $c(t)$ as

$$c_i(t) = c_i + \frac{r_{i,eq}}{\gamma}[1 - \exp(-\gamma t)]. \tag{27}$$

Step II (b): the diffusion effect.

We can get the dynamics of the diffusion effect as

$$\gamma M d\theta = -\frac{\partial L(\theta)}{\partial \theta} dt + [2D(\theta)]^{\frac{1}{2}} dW_t. \tag{28}$$

This is equivalent to SGD dynamics with $\hat{\eta} = \frac{\eta}{\gamma M}$. The expression of $P(\theta, t)$ and $\sigma_i^2(t)$ is directly given by Theorem 1 as

$$\sigma_i^2(t) = \frac{D_i}{\gamma M H_i}[1 - \exp(-\frac{2H_i t}{\gamma M})]. \tag{29}$$

We combine the momentum drift effect and the diffusion effect together, and then obtain the mean squared displacement of $\theta$ as

$$\langle \Delta \theta_i^2(t) \rangle = (c_i(t) - c_i)^2 + \sigma_i^2(t) = \frac{D_i}{\gamma^3 M^2}[1 - \exp(-\gamma t)]^2 + \frac{D_i}{\gamma M H_i}[1 - \exp(-\frac{2H_i t}{\gamma M})]. \tag{30}$$

We respectively introduce the notations of Adam-style Momentum and SGD-style Momentum, and apply the second order Taylor expansion in case of small $-\frac{2H_i t}{\gamma M}$. Then we obtain

$$\langle \Delta \theta_i^2 \rangle = \frac{|H_i| \eta^2}{2(1 - \beta_1)B} \left[1 - \exp\left(-(1 - \beta_1)T\right)\right]^2 + \frac{|H_i| \eta^2 T}{B} + \mathcal{O}(B^{-1} H_i^2 \eta^3 T^2) \quad (31)$$

for Adam-style Momentum, and

$$\langle \Delta \theta_i^2 \rangle = \frac{|H_i| \eta^2}{2(1 - \beta_1)^3 B} \left[1 - \exp\left(-(1 - \beta_1)T\right)\right]^2 + \frac{|H_i| \eta^2 T}{B(1 - \beta_1)^2} + \mathcal{O}(B^{-1} H_i^2 \eta^3 T^2) \quad (32)$$

for SGD-style Momentum.

$\square$

### A.3 Proof of Theorem 3

*Proof.* The proof closely relates to the proof of Theorem 3.2 in Xie et al. (2021b) and a physics work (Kalinay and Percus, 2012).

We first discover how SGD dynamics differs from Momentum dynamics in terms of escaping loss valleys. In this approach, we may transform the proof for Theorem 3.2 of Xie et al. (2021b) into the proof for Theorem 3 with the effective diffusion correction. We use $J$ and $j$ to denote the probability current and the probability flux respectively. According to Gauss Divergence Theorem, we may rewrite the Fokker-Planck Equation (8) as

$$\frac{\partial P(\theta, r, t)}{\partial t} = - r \cdot \nabla_\theta P(\theta, r, t) + \nabla_\theta L(\theta) \cdot M^{-1} \nabla_r P(\theta, r, t)$$
$$+ \nabla_r \cdot M^{-2} D(\theta) \cdot P_{eq}(r) \nabla_r [P_{eq}(r)^{-1} P(\theta, r, t)] \quad (33)$$
$$= - \nabla \cdot J(\theta, t). \quad (34)$$

We will take similar forms of the proof in Xie et al. (2021b) as follows. The mean escape time is written as

$$\tau = \frac{P(\theta \in V_a)}{\int_{S_a} J \cdot dS}. \quad (35)$$

To compute the mean escape time, we decompose the proof into two steps: 1) compute the probability of locating in valley a, $P(\theta \in V_a)$, and 2) compute the probability flux $j = \int_{S_a} J \cdot dS$. The definition of the probability flux integral may refer to Gauss Divergence Theorem.

Step 1:

Fortunately, the stationary probability distribution inside Valley $a$ in Momentum dynamics is also given by a Gaussian distribution in Theorem 1 as

$$\sigma_i^2(t) = \frac{D_i}{\gamma M H_i}. \quad (36)$$

Under Quasi-Equilibrium Assumption, the distribution around minimum a is $P(\theta) = P(a) \exp\left[-\frac{\gamma M}{2} (\theta - a)^\top (D_a^{-\frac{1}{2}} H_a D_a^{-\frac{1}{2}})(\theta - a)\right]$. We use the $T$ notation as the temperature parameter in the stationary distribution, and use the $D$ notation as the diffusion coefficient in the dynamics, for their different roles.

$$P(\theta \in V_a) \quad (37)$$

$$= \int_{\theta \in V_a} P(\theta) dV \quad (38)$$

$$= P(a) \int_{\theta \in V_a} \exp\left[-\frac{\gamma M}{2} (\theta - a)^\top (D_a^{-\frac{1}{2}} H_a D_a^{-\frac{1}{2}})(\theta - a)\right] dV \quad (39)$$

$$\approx P(a) \int_{\theta \in (-\infty, +\infty)} \exp\left[-\frac{\gamma M}{2} (\theta - a)^\top (D_a^{-\frac{1}{2}} H_a D_a^{-\frac{1}{2}})(\theta - a)\right] dV \quad (40)$$

$$= P(a) \frac{(2\pi \gamma M)^{\frac{n}{2}}}{\det(D_a^{-1} H_a)^{\frac{1}{2}}} \quad (41)$$

This result of $P(\theta \in V_a)$ in Momentum only differs from SGD by the temperature correction $\gamma M$.

Step 2: We directly introduce the effective diffusion result from (Kalinay and Percus, 2012) into our analysis. Kalinay and Percus (2012) proved that the phase-space Fokker-Planck Equation (8) can be reduced to a space-dependent Smoluchowski-like equation, which is extended by an effective diffusion correction:

$$\hat{D}_i(\theta) = D_i(\theta) \left( 1 - \sqrt{1 - \frac{4H_i(\theta)}{\gamma^2 M}} \right) \left( \frac{2H_i(\theta)}{\gamma^2 M} \right)^{-1}. \tag{42}$$

As we only employ the Smoluchowski Equation along the escape direction, we use the one-dimensional expression along the escape direction (an eigenvector direction) for simplicity. Without losing clarity, we use the commonly used $T$ to denote the temperature in the proof.

In case of SGD (Xie et al., 2021b), we can obtain Smoluchowski Equation in position space:

$$J = D(\theta) \exp\left( \frac{-L(\theta)}{T} \right) \nabla \left[ \exp\left( \frac{L(\theta)}{T} \right) P(\theta) \right], \tag{43}$$

where $T = D$. According to (Kalinay and Percus, 2012), in case of finite inertia, we can transform the phase-space equation into the position-space Smoluchowski-like form with the effective diffusion correction:

$$J = \hat{D}(\theta) \exp\left( \frac{-L(\theta)}{T} \right) \nabla \left[ \exp\left( \frac{L(\theta)}{T} \right) P(\theta) \right], \tag{44}$$

where $T = \frac{D}{\gamma M}$ , and $\hat{D}$ defined by Equation (42) replaces standard $D$.

We assume the point $s$ is a midpoint on the most possible path between a and b, where $L(s) = (1-s)L(a) + sL(b)$. The temperature $T_a$ dominates the path $a \to s$, while temperature $T_b$ dominates the path $s \to b$. So we have

$$\nabla \left[ \exp\left( \frac{L(\theta) - L(s)}{T} \right) P(\theta) \right] = JD^{-1} \exp\left( \frac{L(\theta) - L(s)}{T} \right). \tag{45}$$

We integrate the equation from Valley a to the outside of Valley a along the most possible escape path

$$Left = \int_a^c \frac{\partial}{\partial \theta} [\exp\left( \frac{L(\theta) - L(s)}{T} \right) P(\theta)] d\theta \tag{46}$$

$$= \int_a^s \frac{\partial}{\partial \theta} \left[ \exp\left( \frac{L(\theta) - L(s)}{T_a} \right) P(\theta) \right] d\theta \tag{47}$$

$$+ \int_s^c \frac{\partial}{\partial \theta} \left[ \exp\left( \frac{L(\theta) - L(s)}{T_b} \right) P(\theta) \right] d\theta \tag{48}$$

$$= [P(s) - \exp\left( \frac{L(a) - L(s)}{T_a} \right) P(a)] + [0 - P(s)] \tag{49}$$

$$= - \exp\left( \frac{L(a) - L(s)}{T_a} \right) P(a) \tag{50}$$

$$Right = - J \int_a^c D^{-1} \exp\left( \frac{L(\theta) - L(s)}{T} \right) d\theta \tag{51}$$

We move $J$ to the outside of integral based on Gauss's Divergence Theorem, because $J$ is fixed on the escape path from one minimum to another. As there is no field source on the escape path, $\int_V \nabla \cdot J(\theta) dV = 0$ and $\nabla J(\theta) = 0$. Obviously, probability sources are all near minima in deep learning. So we obtain

$$J = \frac{\exp\left( \frac{L(a) - L(s)}{T_a} \right) P(a)}{\int_a^c \hat{D}^{-1} \exp\left( \frac{L(\theta) - L(s)}{T} \right) d\theta}. \tag{52}$$

Near saddle points, we have

$$\int_a^c \hat{D}^{-1} \exp\left(\frac{L(\theta) - L(s)}{T}\right) d\theta \tag{53}$$

$$\approx \int_a^c \hat{D}^{-1} \exp\left[\frac{L(b) - L(s) + \frac{1}{2}(\theta - b)^\top H_b(\theta - b)}{T_b}\right] d\theta \tag{54}$$

$$\approx \hat{D}_b^{-1} \int_{-\infty}^{+\infty} \exp\left[\frac{L(b) - L(s) + \frac{1}{2}(\theta - b)^\top H_b(\theta - b)}{T_b}\right] d\theta \tag{55}$$

$$= \hat{D}_b^{-1} \exp\left(\frac{L(b) - L(s)}{T_b}\right) \sqrt{\frac{2\pi T_b}{|H_b|}}. \tag{56}$$

Besides the temperature correction $T = \frac{D}{\gamma M}$, this result of $J$ in Momentum also differs from SGD by the effective diffusion correction $\frac{\hat{D}_b}{D_b}$. The effective diffusion correction coefficient is given by

$$\frac{\hat{D}_i(\theta)}{D_i(\theta)} = \frac{1 - \sqrt{1 - \frac{4H_i(\theta)}{\gamma^2 M}}}{\frac{2H_i(\theta)}{\gamma^2 M}}. \tag{57}$$

Based on the formula of the one-dimensional probability current and flux, we obtain the high-dimensional flux escaping through b:

$$\int_{S_b} J \cdot dS \tag{58}$$

$$= J_{1d} \int_{S_b} \exp\left[-\frac{\gamma M}{2}(\theta - b)^\top [D_b^{-\frac{1}{2}} H_b D_b^{-\frac{1}{2}}]^{\perp e}(\theta - b)\right] dS \tag{59}$$

$$= J_{1d} \frac{(2\pi\gamma M)^{\frac{n-1}{2}}}{(\prod_{i \neq e}(D_{bi}^{-1} H_{bi}))^{\frac{1}{2}}} \tag{60}$$

$$= \frac{\exp\left(\frac{L(a) - L(s)}{T_{ae}}\right) P(a) \frac{(2\pi\gamma M)^{\frac{n-1}{2}}}{(\prod_{i \neq e}(D_{bi}^{-1} H_{bi}))^{\frac{1}{2}}}}{\hat{D}_{be}^{-1} \exp\left(\frac{L(b) - L(s)}{T_{be}}\right) \sqrt{\frac{2\pi T_{be}}{|H_{be}|}}} \tag{61}$$

where $[\cdot]^{\perp e}$ indicates the directions perpendicular to the escape direction $e$.

Based on the results of Step 1 and Step 2, we have

$$\tau = \frac{P(\theta \in V_a)}{\int_{S_b} J \cdot dS} \tag{62}$$

$$= P(a) \frac{(2\pi\gamma M)^{\frac{n}{2}}}{\det(D_a^{-1} H_a)^{\frac{1}{2}}} \frac{\hat{D}_{be}^{-1} \exp\left(\frac{L(b) - L(s)}{T_{be}}\right) \sqrt{\frac{2\pi T_{be}}{|H_{be}|}}}{\exp\left(\frac{L(a) - L(s)}{T_{ae}}\right) P(a) \frac{(2\pi\gamma M)^{\frac{n-1}{2}}}{(\prod_{i \neq e}(D_{bi}^{-1} H_{bi}))^{\frac{1}{2}}}} \tag{63}$$

$$= \frac{1}{\hat{D}_{be}} 2\pi \frac{D_{be}}{|H_{be}|} \exp\left[\frac{2\gamma M B \Delta L}{\eta}\left(\frac{s}{H_{ae}} + \frac{(1-s)}{|H_{be}|}\right)\right] \tag{64}$$

$$= \pi \left[\sqrt{1 + \frac{4|H_{be}|}{\gamma^2 M}} + 1\right] \frac{1}{|H_{be}|} \exp\left[\frac{2\gamma M B \Delta L}{\eta}\left(\frac{s}{H_{ae}} + \frac{(1-s)}{|H_{be}|}\right)\right]. \tag{65}$$

We have replaced the eigenvalue of $H_b$ along the escape direction by its absolute value.

Finally, by introducing $\gamma$ and $M$, we obtain the log-scale expression as

$$\log(\tau) = \mathcal{O}\left(\frac{2(1 - \beta_1) B \Delta L}{\beta_3 \eta H_{ae}}\right) \tag{66}$$

$\square$

## A.4 PROOF OF PROPOSITION 2

*Proof.* The proof closely relates to the proof of 3. We only need replace the standard learning rate by the adaptive learning rate $\hat{\eta} = \eta v^{-\frac{1}{2}}$, and set $\gamma M = 1$. Particularly,

$$D_{adam} = Dv^{-\frac{1}{2}} = \frac{\eta[H]^+ v^{-\frac{1}{2}}}{2B} = \frac{1}{2}\eta\sqrt{\frac{[H]^+}{B}}. \tag{67}$$

We introduce $\hat{\eta} = \eta v^{-\frac{1}{2}}$ into the proof of 3, and obtain

$$\tau = \frac{P(\theta \in V_a)}{\int_{S_b} J \cdot dS} \tag{68}$$

$$= \left[\frac{|\det(D_b^{-1} V_b^{\frac{1}{2}} H_b)|}{\det(D_a^{-1} V_a^{\frac{1}{2}} H_a)}\right]^{\frac{1}{2}} \pi \left[\sqrt{1 + \frac{4|H_{be}|}{\gamma^2 M}} + 1\right] \frac{1}{|H_{be}|} \exp\left[\frac{2\gamma MB\Delta L}{\eta}\left(\frac{s}{V_{ae}^{-\frac{1}{2}} H_{ae}} + \frac{(1-s)}{V_{be}^{-\frac{1}{2}}|H_{be}|}\right)\right] \tag{69}$$

$$= \pi \left[\sqrt{1 + \frac{4\eta\sqrt{B|H_{be}|}}{1 - \beta_1}} + 1\right] \frac{|\det(H_a^{-1} H_b)|^{\frac{1}{4}}}{|H_{be}|} \exp\left[\frac{2\sqrt{B}\Delta L}{\eta}\left(\frac{s}{\sqrt{H_{ae}}} + \frac{(1-s)}{\sqrt{|H_{be}|}}\right)\right]. \tag{70}$$

$\square$

## A.5 PROOF OF PROPOSITION 4

*Proof.* The proof closely relates to the proof of 3. We only need introduce the mass matrix $M$ and the dampening matrix $\gamma$. Fortunately, $\gamma M = I$. Thus we directly have the result from the proof of 3 as:

$$\tau = \frac{P(\theta \in V_a)}{\int_{S_b} J \cdot dS} \tag{71}$$

$$= \pi \left[\sqrt{1 + \frac{4|H_{be}|}{\gamma^2 M}} + 1\right] \frac{1}{|H_{be}|} \exp\left[\frac{2\gamma MB\Delta L}{\eta}\left(\frac{s}{H_{ae}} + \frac{(1-s)}{|H_{be}|}\right)\right]. \tag{72}$$

As $M = \eta(I - \beta_1)^{-1}$, $\gamma = \eta^{-1}(I - \beta_1)$, and $I - \beta_1 = \frac{\beta_0}{v}v$, we obtain the result:

$$\tau = \pi \left[\sqrt{1 + \frac{4\eta\sum_{i=1}^n |H_{bi}|}{\beta_0 n}} + 1\right] \frac{1}{|H_{be}|} \exp\left[\frac{2B\Delta L}{\eta}\left(\frac{s}{H_{ae}} + \frac{(1-s)}{|H_{be}|}\right)\right] \tag{73}$$

$\square$

## A.6 PROOF OF THEOREM 4

Without loss of generality, we assume the dimensionality is one and rewrite the main updating rule of Adai as

$$\theta_{t+1} = \theta_t - \eta(1 - \beta_{1,t})g_t + \beta_{1,t}(\theta_t - \theta_{t-1}). \tag{74}$$

In the convergence proof, we do not need to specify how to update $\beta_{1,t}$ but just let $\beta_{1,t} \in [0, 1)$. We denote that $\theta_{-1} = \theta_0$.

Before presenting the main proof, we first prove four useful lemmas.

**Lemma 1.** *Under the conditions of Theorem 4, for any $t \geq 0$, we have*

$$x_{t+1} - x_t = -\eta \sum_{k=0}^t q_{k,t} g_t,$$

*where*

$$q_{k,t} = (1 - \beta_k) \prod_{i=k+1}^{t} \beta_{1,i}.$$

*Then we have*

$$1 - \beta_{1,\max}^{t+1} \leq \sum_{k=0}^{t} q_{k,t} \leq 1.$$

*Proof.* Recall that

$$\theta_{t+1} = \theta_t - \eta(1 - \beta_{1,t})g_t + \beta_{1,t}(\theta_t - \theta_{t-1}) \tag{75}$$

$$\theta_{t+1} - \theta_t = \beta_{1,t}(\theta_t - \theta_{t-1}) - \eta(1 - \beta_{1,t})g_t. \tag{76}$$

Then we have

$$\theta_{t+1} - \theta_t = -\eta \sum_{k=0}^{t} (1 - \beta_{1,k})g_k \prod_{i=k+1}^{t} \beta_{1,i}. \tag{77}$$

Let $q_{k,t} = (1 - \beta_k) \prod_{i=k+1}^{t} \beta_{1,i}$.

For analyzing the maximum and the minimum, we calculate the derivatives with respect to $\beta_{1,k}$ for any $0 \leq k \leq t$:

$$\frac{\partial \sum_{k=0}^{t} q_{k,t}}{\partial \beta_{1,0}} = -\prod_{i=k+1}^{t} \beta_{1,i} \leq 0. \tag{78}$$

Note that $0 \leq \beta_{1,k} \leq \beta_{1,\max}$. Then we have

$$\sum_{k=0}^{t} q_{k,t}|_{\beta_{1,0}=\beta_{1,\max}} \leq \sum_{k=0}^{t} q_{k,t} \leq \sum_{k=0}^{t} q_{k,t}|_{\beta_{1,0}=0}. \tag{79}$$

Recursively, we can calculate the derivatives with respect to $\beta_{1,1}, \beta_{1,2}, \ldots, \beta_{1,t}$.

Then we obtain $\max(\sum_{k=0}^{t} q_{k,t}) = 1$ by letting $\beta_{1,k} = 0$ for all $k$.

Similarly, we obtain $\min(\sum_{k=0}^{t} q_{k,t}) = 1$ by letting $\beta_{1,t} = \beta_{1,\max}$ for all $k$.

Then we obtain

$$1 - \beta_{1,\max}^{t+1} \leq \sum_{k=0}^{t} q_{k,t} \leq 1 \tag{80}$$

The proof is now complete. $\qquad\square$

**Lemma 2.** *Under the conditions of Theorem 4, for any $t \geq 0$, we have*

$$\mathbb{E}[L(\theta_{t+1}) - L(\theta_t)] \leq \frac{1}{2} \sum_{k=0}^{t} q_{k,t}\mathbb{E}[\sum_{j=k}^{t-1} \|\nabla L(\theta_{j+1}) - L(\theta_j)\|^2] +$$

$$\sum_{k=0}^{t} q_{k,t}(\frac{t-k}{2}\eta^2 - \eta)\mathbb{E}[\|\nabla L(\theta_k)\|^2] + \frac{\mathcal{L}\eta^2}{2}\|\sum_{k=0}^{t} q_{k,t}g_k\|^2.$$

*Proof.* As $L(\theta)$ is $\mathcal{L}$-smooth, we have

$$L(\theta_{t+1}) - L(\theta_t) \tag{81}$$

$$\leq \nabla L(\theta_t)^\top (\theta_{t+1} - \theta_t) + \frac{\mathcal{L}}{2} \|\theta_{t+1} - \theta_t\|^2 \tag{82}$$

$$= -\eta \sum_{k=0}^{t} q_{k,t} \nabla L(\theta_t)^\top g_k + \frac{\mathcal{L}\eta^2}{2} \|\sum_{k=0}^{t} q_{k,t} g_k\|^2 \tag{83}$$

$$= -\eta \sum_{k=0}^{t} q_{k,t} \sum_{j=k}^{t-1} (\nabla L(\theta_{j+1}) - \nabla L(\theta_j))^\top (\nabla L(\theta_k) + \xi_k) - $$

$$\eta \sum_{k=0}^{t} q_{k,t} \nabla L(\theta_k)^\top (\nabla L(\theta_k) + \xi_k) + \frac{\mathcal{L}\eta^2}{2} \|\sum_{k=0}^{t} q_{k,t} g_k\|^2. \tag{84}$$

Taking expectation on both sides gives

$$\mathbb{E}[L(\theta_{t+1}) - L(\theta_t)] \tag{85}$$

$$\leq \frac{1}{2\mathcal{L}} \sum_{k=0}^{t} q_{k,t} \mathbb{E}[\sum_{j=k}^{t-1} (\nabla L(\theta_{j+1}) - \nabla L(\theta_j))^\top \nabla L(\theta_k)] - \tag{86}$$

$$\sum_{k=0}^{t} q_{k,t} \eta \mathbb{E}[\|\nabla L(\theta_k)\|^2] + \frac{\mathcal{L}\eta^2}{2} \|\sum_{k=0}^{t} q_{k,t} g_k\|^2. \tag{87}$$

By the Cauchy-Schwarz Inequality, we have

$$\mathbb{E}[L(\theta_{t+1}) - L(\theta_t)] \tag{88}$$

$$\leq \sum_{k=0}^{t} q_{k,t} \mathbb{E}[\frac{1}{2} \|\nabla L(\theta_t) - \nabla L(\theta_k)\|^2 + \frac{\eta^2}{2} \|\nabla L(\theta_k)\|^2] - $$

$$\eta \sum_{k=0}^{t} q_{k,t} \mathbb{E}[\|\nabla L(\theta_k)\|^2] + \frac{\mathcal{L}\eta^2}{2} \mathbb{E}[\|\sum_{k=0}^{t} q_{k,t} g_k\|^2] \tag{89}$$

$$= \frac{1}{2} \sum_{k=0}^{t} q_{k,t} \mathbb{E}[\sum_{j=k}^{t-1} \|\nabla L(\theta_{j+1}) - L(\theta_j)\|^2] + $$

$$\sum_{k=0}^{t} q_{k,t} (\frac{t-k}{2} \eta^2 - \eta) \mathbb{E}[\|\nabla L(\theta_k)\|^2] + \frac{\mathcal{L}\eta^2}{2} \|\sum_{k=0}^{t} q_{k,t} g_k\|^2. \tag{90}$$

The proof is now complete. $\qquad\square$

**Lemma 3.** *Under the conditions of Theorem 4, for any $t \geq 0$, we have*

$$\sum_{k=0}^{t} q_{k,t}(t-k) \leq \frac{\beta_{1,\max}}{1 - \beta_{1,\max}}$$

*Proof.* For analyzing the maximum, we calculate the derivatives with respect to $\beta_{1,k}$ for any $0 \leq k \leq t$.

With respect to $\beta_{1,1}$, we have

$$\frac{\partial \sum_{k=0}^{t} q_{k,t}(t-k)}{\partial \beta_{1,0}} = -\prod_{k=1}^{t} \beta_{1,k} t \leq 0. \tag{91}$$

Then we let $\beta_{1,0} = 0$ and obtain the derivative with respect to $\beta_{1,1}$ as

$$\frac{\partial \sum_{k=0}^{t} q_{k,t}(t-k)}{\partial \beta_{1,1}} = \prod_{k=2}^{t} \beta_{1,k}(t-(t-1)) \geq 0. \tag{92}$$

Then we let $\beta_{1,j} = \beta_{1,\max}$ for $1 \leq j \leq k-1$ and obtain the derivative with respect to $\beta_{1,k}$ as

$$\frac{\partial \sum_{k=0}^{t} q_{k,t}(t-k)}{\partial \beta_{1,k}} \geq (t-k)\frac{\partial \sum_{k=0}^{t} q_{k,t}}{\partial \beta_{1,k}} = 0. \tag{93}$$

Thus, we have $\beta_{1,0} = 0$ and $\beta_{1,k} = \beta_{1,\max}$ for $1 \leq j \leq t$ to maximize $\sum_{k=0}^{t-1} q_{k,t}(t-k)$. We may write it as

$$\sum_{k=0}^{t} q_{k,t}(t-k) \leq \beta_{1,\max}^{t+1}t + \sum_{k=0}^{t}(1-\beta_{1,\max})\beta_{1,\max}^{k}k \tag{94}$$

Note that $(1-\beta_{1,\max})\beta_{1,\max}^{k}k$ is an arithmetico-geometric sequence. Thus, we have

$$\max(\sum_{k=0}^{t} q_{k,t}(t-k)) \leq \beta_{1,\max}^{t+1}t + \left[\beta_{1,\max} - \beta_{1,\max}^{t+1}t + \frac{\beta_{1,\max}^{2}(1-\beta_{1,\max}^{t-1})}{1-\beta_{1,\max}}\right] \tag{95}$$

$$\leq \beta_{1,\max} + \frac{\beta_{1,\max}^{2}}{1-\beta_{1,\max}} \tag{96}$$

$$= \frac{\beta_{1,\max}}{1-\beta_{1,\max}} \tag{97}$$

The proof is now complete. $\qquad\square$

**Lemma 4.** *Under the conditions of Theorem 4, for any $t \geq 0$, we have*

$$\frac{1}{2}\sum_{k=0}^{t} q_{k,t}\mathbb{E}[\sum_{j=k}^{t-1}\|\nabla L(\theta_{j+1}) - L(\theta_j)\|^2] + \frac{\mathcal{L}\eta^2}{2}\|\sum_{k=0}^{t} q_{k,t}g_k\|^2 \leq \frac{L\eta^2}{2(1-\beta_{1,\max})}(G^2 + \delta^2).$$

*Proof.* As $L(\theta)$ is $\mathcal{L}$-smooth, we have

$$\|\nabla L(\theta_{j+1}) - L(\theta_j)\|^2 \tag{98}$$

$$\leq \mathcal{L}\|\theta_{j+1} - \theta_j\|^2 \tag{99}$$

$$= \mathcal{L}\| - \eta\sum_{i=0}^{j} q_{i,j}g_i\|^2. \tag{100}$$

By $\sum_{i=0}^{j} q_{i,j} \leq 1$ in Lemma 1, we have

$$\mathbb{E}[\|\nabla L(\theta_{j+1}) - L(\theta_j)\|^2] \tag{101}$$

$$\leq \mathcal{L}\eta^2(G^2 + \delta^2). \tag{102}$$

and

$$\frac{\mathcal{L}\eta^2}{2}\|\sum_{k=0}^{t} q_{k,t}g_k\|^2 \tag{103}$$

$$\leq \frac{\mathcal{L}\eta^2}{2}(G^2 + \delta^2). \tag{104}$$

By Lemma 3, we have

$$\frac{1}{2}\sum_{k=0}^{t} q_{k,t}\mathbb{E}[\sum_{j=k}^{t-1}\|\nabla L(\theta_{j+1}) - L(\theta_j)\|^2] \tag{105}$$

$$\leq \frac{\eta^2}{2}(G^2 + \delta^2)\sum_{k=0}^{t} q_{k,t}(t-k) \tag{106}$$

$$\leq \frac{\eta^2\beta_{1,\max}}{2(1-\beta_{1,\max})}(G^2 + \delta^2). \tag{107}$$

Then we obtain

$$\frac{1}{2}\sum_{k=0}^{t}q_{k,t}\mathbb{E}[\sum_{j=k}^{t-1}\|\nabla L(\theta_{j+1}) - L(\theta_j)\|^2] + \frac{\mathcal{L}\eta^2}{2}\|\sum_{k=0}^{t}q_{k,t}g_k\|^2 \tag{108}$$

$$\leq \frac{\mathcal{L}\eta^2\beta_{1,\max}}{2(1-\beta_{1,\max})}(G^2+\delta^2) + \frac{L\eta^2}{2}(G^2+\delta^2) \tag{109}$$

$$\leq \frac{\mathcal{L}\eta^2}{2(1-\beta_{1,\max})}(G^2+\delta^2). \tag{110}$$

$\square$

*Proof.* The proof of Theorem 4 is organized as follows.

By Lemma 2 and Lemma 4, we have

$$\mathbb{E}[L(\theta_{t+1}) - L(\theta_t)]$$

$$\leq \frac{1}{2}\sum_{k=0}^{t}q_{k,t}\mathbb{E}[\sum_{j=k}^{t-1}\|\nabla L(\theta_{j+1}) - L(\theta_j)\|^2]+$$

$$\sum_{k=0}^{t}q_{k,t}(\frac{t-k}{2}\eta^2 - \eta)\mathbb{E}[\|\nabla L(\theta_k)\|^2] + \frac{\mathcal{L}\eta^2}{2}\|\sum_{k=0}^{t}q_{k,t}g_k\|^2 \tag{111}$$

$$\leq \sum_{k=0}^{t}q_{k,t}(\frac{t-k}{2}\eta^2 - \eta)\mathbb{E}[\|\nabla L(\theta_k)\|^2] + \frac{\mathcal{L}\eta^2}{2(1-\beta_{1,\max})}(G^2+\delta^2). \tag{112}$$

Thus, we have

$$\sum_{k=0}^{t}q_{k,t}\eta\mathbb{E}[\|\nabla L(\theta_k)\|^2]$$

$$\leq \mathbb{E}[L(\theta_t) - L(\theta_{t+1})] + \sum_{k=0}^{t}q_{k,t}\frac{t-k}{2}\eta^2\mathbb{E}[\|\nabla L(\theta_k)\|^2] + \frac{\mathcal{L}\eta^2}{2(1-\beta_{1,\max})}(G^2+\delta^2). \tag{113}$$

By Lemma 1 and Lemma 3, we have

$$(1-\beta_{1,\max}^{t+1})\eta\min_{k=0,\ldots,t}\mathbb{E}[\|\nabla L(\theta_k)\|^2]$$

$$\leq \mathbb{E}[L(\theta_t) - L(\theta_{t+1})] + \frac{\beta_{1,\max}\eta^2}{2(1-\beta_{1,\max})}G^2 + \frac{\mathcal{L}\eta^2}{2(1-\beta_{1,\max})}(G^2+\delta^2). \tag{114}$$

By summing the above inequality for $t = 0, \ldots, t$, we have

$$(t+1)(1-\beta_{1,\max})\eta\min_{k=0,\ldots,t}\mathbb{E}[\|\nabla L(\theta_k)\|^2]$$

$$\leq L(\theta_0) - L^\star + \frac{\beta_{1,\max}\eta^2}{2(1-\beta_{1,\max})}G^2(t+1) + \frac{\mathcal{L}\eta^2}{2(1-\beta_{1,\max})}(G^2+\delta^2)(t+1). \tag{115}$$

Then

$$\min_{k=0,\ldots,t}\mathbb{E}[\|\nabla L(\theta_k)\|^2]$$

$$\leq \frac{L(\theta_0) - L^\star}{(1-\beta_{1,\max})(t+1)\eta} + \frac{\beta_{1,\max}\eta}{2(1-\beta_{1,\max})^2}G^2 + \frac{\mathcal{L}\eta}{2(1-\beta_{1,\max})^2}(G^2+\delta^2). \tag{116}$$

Let $\eta \leq \frac{C}{\sqrt{t+1}}$. We have

$$
\min_{k=0,\ldots,t} \mathbb{E}[\|\nabla L(\theta_k)\|^2]
$$

$$
\leq \frac{L(\theta_0) - L^\star}{(1 - \beta_{1,\max})C\sqrt{t+1}} + \frac{\beta_{1,\max}C}{2(1 - \beta_{1,\max})^2\sqrt{t+1}}G^2 +
$$

$$
\frac{\mathcal{L}C}{2(1 - \beta_{1,\max})^2\sqrt{t+1}}(G^2 + \delta^2) \tag{117}
$$

$$
\tag{118}
$$

The proof is now complete. $\qquad\square$

## B  CLASSICAL APPROXIMATION ASSUMPTIONS

Assumption 2 indicates that the dynamical system is in equilibrium near minima but not necessarily near saddle points. It means that $\frac{\partial P(\theta,t)}{\partial t} \approx 0$ holds near minima, but not necessarily holds near saddle point $b$. Quasi-Equilibrium Assumption is actually weaker but more useful than the conventional stationary assumption for deep learning (Welling and Teh, 2011; Mandt et al., 2017). Under Assumption 2, the probability density $P$ can behave like a stationary distribution only inside valleys, but density transportation through saddle points can be busy. Quasi-Equilibrium is more like: stable lakes (loss valleys) is connected by rapid Rivers (escape paths). In contrast, the stationary assumption requires strictly zero flux between lakes (loss valleys). Little knowledge about density motion can be obtained under the stationary assumption.

Low Temperature Assumption is common (Van Kampen, 1992; Zhou, 2010; Berglund, 2013; Jastrzkebski et al., 2017), and is always justified when $\frac{\eta}{B}$ is small. Under Assumption 3, the probability densities will concentrate around minima and MPPs. Numerically, the 6-sigma rule may often provide good approximation for a Gaussian distribution. Assumption 3 will make the second order Taylor approximation, Assumption 1, even more reasonable in SGD diffusion.

Here, we try to provide a more intuitive explanation about Low Temperature Assumption in the domain of deep learning. Without loss of generality, we discuss it in one-dimensional dynamics. The temperature can be interpreted as a real number $D$. In SGD, we have the temperature as $D = \frac{\eta}{2B}H$. In statistical physics, if $\frac{\Delta L}{D}$ is large, then we call it Low Temperature Approximation. Note that $\frac{\Delta L}{D}$ appears insides an exponential function in the theoretical analysis. People usually believe that, numerically, $\frac{\Delta L}{D} > 6$ can make a good approximation, for a similar reason of the 6-sigma rule in statistics. In the final training phase of deep networks, a common setting is $\eta = 0.01$ and $B = 128$. Thus, we may safely apply Assumption 3 to the loss valleys which satisfy the very mild condition $\frac{\Delta L}{H} > 2.3 \times 10^{-4}$. Empirically, the condition $\frac{\Delta L}{H} > 2.3 \times 10^{-4}$ holds well in SGD dynamics.

## C  STOCHASTIC GRADIENT NOISE ANALYSIS

In this section, we empirically discussed the covariance of stochastic gradient noise (SGN) and why SGN is approximately Gaussian in common settings.

In Figure 6, following Xie et al. (2021b), we again validated the relation between gradient noise covariance and the Hessian. We particularly choose a randomly initialized mode so that the model is not near critical points. We display all elements $H_{(i,j)} \in [1e-4, 0.5]$ of the Hessian matrix and the corresponding elements $C_{(i,j)}$ of gradient covariance matrix in the space spanned by the eigenvectors of Hessian. Even if the model is far from critical points, the SGN covariance is still approximately proportional to the Hessian and inverse to the batch size $B$. The correlation is especially high along the directions with small-magnitude eigenvalues of the Hessian. Fortunately, small-magnitude eigenvalues of the Hessian indicate the flat directions which we care most in saddle-point escaping.

We also note that the SGN we study is introduced by minibatch training, $\frac{\partial L(\theta_t)}{\partial \theta_t} - \frac{\partial \hat{L}(\theta_t)}{\partial \theta_t}$, which is the difference between gradient descent and stochastic gradient descent.

In Figure 7 of the Appendix, Xie et al. (2021b) empirically verified that SGN is highly similar to Gaussian noise instead of heavy-tailed Lévy noise. Xie et al. (2021b) recovered the experiment of

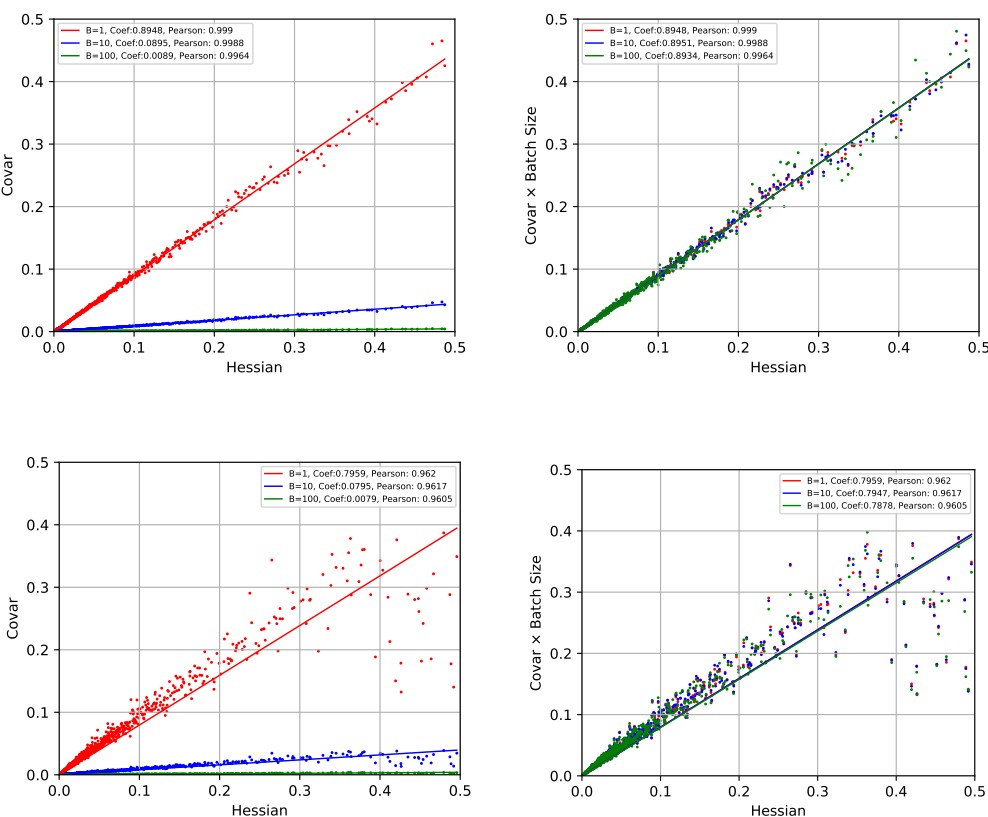

Figure 6: We verified Equation (3) by training pretrained and random three-layer fully-connected networks on MNIST (LeCun, 1998). Top Row: Pretrained Models. Bottom Row: Random Models. The noise covariance is approximately proportional to the Hessian and inverse to the batch size $B$ even not around critical points.

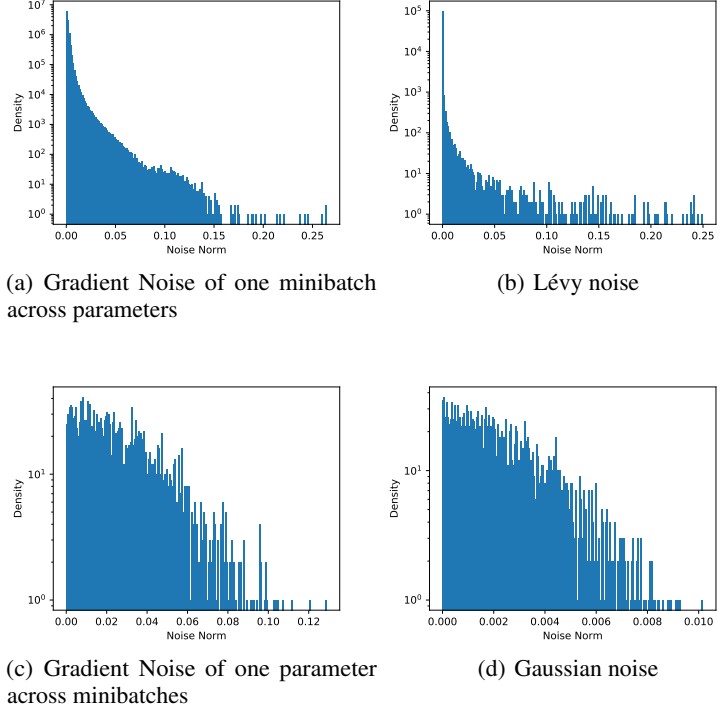

(a) Gradient Noise of one minibatch across parameters

(b) Lévy noise

(c) Gradient Noise of one parameter across minibatches

(d) Gaussian noise

Figure 7: The Stochastic Gradient Noise Analysis (Xie et al., 2021b). The histogram of the norm of the gradient noises computed with ResNet18 on CIFAR-10. Subfigure (a) follows Simsekli et al. (2019) and computes "stochastic gradient noise" across parameters. Subfigure (c) follows the usual definition and computes stochastic gradient noise across minibatches. Obviously, SGN computed over minibatches is more like Gaussian noise rather than Lévy noise.

Simsekli et al. (2019) to show that gradient noise is approximately Lévy noise only if it is computed across parameters. Figure 7 of the Appendix actually suggests that the contradicted observations are from the different formulations of gradient noise. Simsekli et al. (2019) computed "SGN" across $n$ model parameters and regarded "SGN" as $n$ samples drawn from a *single-variant* distribution. In Xie et al. (2021b), SGN computed over different minibatches obeys a $N$-*variant* Gaussian distribution, which can be $\theta$-*dependent and anisotropic*. Simsekli et al. (2019) studied the distribution of SGN as a single-variant distribution, while Xie et al. (2021) relaxed it as a $n$-variant distribution. Figure 7 holds well at least when the batch size $B$ is larger than 16, which is common in practice.

## D EXPERIMENTAL DETAILS

**Computational environment.** The experiments are conducted on a computing cluster with GPUs of NVIDIA® Tesla™ P100 16GB and CPUs of Intel® Xeon® CPU E5-2640 v3 @ 2.60GHz.

### D.1 IMAGE CLASSIFICATION

**Data Preprocessing:** For CIFAR-10/CIFAR-100, we perform the per-pixel zero-mean unit-variance normalization, horizontal random flip, and $32 \times 32$ random crops after padding with 4 pixels on each side. For ImageNet, we perform the per-pixel zero-mean unit-variance normalization, horizontal random flip, and the resized random crops where the random size (of 0.08 to 1.0) of the original size and a random aspect ratio (of $\frac{3}{4}$ to $\frac{4}{3}$) of the original aspect ratio is made.

**Hyperparameter Settings for CIFAR-10 and CIFAR-100:** We select the optimal learning rate for each experiment from $\{0.00001, 0.0001, 0.001, 0.01, 0.1, 1, 10\}$ for non-adaptive gradient methods and use the default learning rate in original papers for adaptive gradient methods. The settings of

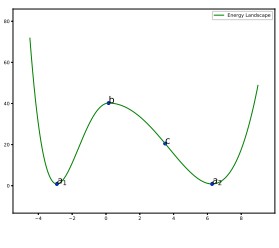 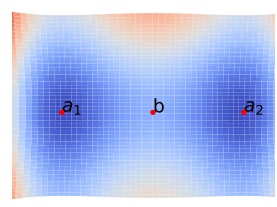

(a) 1-Dimensional Escape      (b) High-Dimensional Escape

Figure 8: The illustration of Kramers Escape Problems (Xie et al., 2021b). Assume there are two valleys, Sharp Valley $a_1$ and Flat Valley $a_2$. Also Col b is the boundary between two valleys. $a_1$ and $a_a$ are minima of two neighboring valleys. $b$ is the saddle point separating the two valleys. $c$ locates outside of Valley $a_1$.

learning rates: $\eta = 1$ for Adai;$\eta = 0.1$ for SGD with Momentum and AdaiW; $\eta = 0.001$ for Adam, AMSGrad, AdamW, AdaBound, Yogi, and RAdam; $\eta = 0.01$ for Padam. For the learning rate schedule, the learning rate is divided by 10 at the epoch of $\{80, 160\}$ for CIFAR-10 and $\{100, 150\}$ for CIFAR-100. The batch size is set to 128 for CIFAR-10 and CIFAR-100. The $L_2$ regularization hyperparameter is set to $\lambda = 0.0005$ for CIFAR-10 and CIFAR-100. Considering the linear scaling rule of decoupled weight decay and initial learning rates (Loshchilov and Hutter, 2018), we chose decoupled weight decay hyperparameters as: $\lambda = 0.5$ for AdamW on CIFAR-10 and CIFAR-100; $\lambda = 0.005$ for AdaiW on CIFAR-10 and CIFAR-100. We set the momentum hyperparameter $\beta_1 = 0.9$ for SGD with Momentum. As for other optimizer hyperparameters, we apply the default hyperparameter settings directly.

**Hyperparameter Settings for ImageNet:** We select the optimal learning rate for each experiment from $\{0.00001, 0.0001, 0.001, 0.01, 0.1, 1, 10\}$ for Adai, SGD with Momentum, and Adam. The settings of learning rates: $\eta = 1$ for Adai;$\eta = 0.1$ for SGD with Momentum; $\eta = 0.0001$ for Adam. For the learning rate schedule, the learning rate is divided by 10 at the epoch of $\{30, 60, 90\}$. The batch size is set to 256. The $L_2$ regularization hyperparameter is set to $\lambda = 0.0001$. We set the momentum hyperparameter $\beta_1 = 0.9$ for SGD with Momentum. As for other training hyperparameters, we apply the default hyperparameter settings directly.

Some papers often choose $\lambda = 0.0001$ as the default weight decay setting for CIFAR-10 and CIFAR-100. We study the weight decay setting in Appendix F.

### D.2 Language Modeling

Moreover, we present the learning curves for language modeling experiments in Figure 12. We empirically compare three base optimizers, including Adai, SGD with Momentum, and Adam, for language modeling experiments. We use a classical language model, Long Short-Term Memory (LSTM) (Hochreiter and Schmidhuber, 1997b) with 2 layers, 512 embedding dimensions, and 512 hidden dimensions, which has 14 million model parameters and is similar to "medium LSTM" in Zaremba et al. (2014). The benchmark task is the word-level Penn TreeBank (Marcus et al., 1993).

**Hyperparameter Settings.** Batch Size: $B = 20$. BPTT Size: $bptt = 35$. Weight Decay: $\lambda = 0.00005$. Learning Rate: $\eta = 0.001$. The dropout probability is set to 0.5. We clipped gradient norm to 1. We select optimal learning rates from $\{10, 1, 0.1, 0.01, 0.001, 0.0001, 0.00001\}$ for each optimizer.

## E    The Mean Escape Time Analysis

**Mean Escape Time:** The mean escape time is the expected time for a particle governed by Equation 1 to escape from Sharp Valley $a_1$ to Flat Valley $a_2$, seen in Figure 8. The mean escape time is widely used in related statistical physics and stochastic process (Van Kampen, 1992; Nguyen et al., 2019).

Related machine learning papers (Xie et al., 2021b; Hu et al., 2019; Nguyen et al., 2019) also studied how SGD selects minima by using the concept of the mean escape time.

**Data Set:** We generate 50000 Gaussian samples as the training data set, where $x \sim \mathcal{N}(0, 4I)$.

**Hyperparameters**: The batch size is set 10. No weight decay. The learning rate: 0.001 for Adai, 0.0001 for Momentum (with $\beta = 0.9$), and 0.03 for Adam.

**Test Function:** Styblinski-Tang Function is a commonly used function in nonconvex optimization, written as

$$f(\theta) = \frac{1}{2} \sum_{i=1}^{N} (\theta_i^4 - 16\theta_i^2 + 5\theta_i).$$

We use 10-dimensional Styblinski-Tang Function as the test function, and Gaussian samples as training data.

$$L(\theta) = f(\theta - x),$$

where data samples $x \sim \mathcal{N}(0, 4I)$. The one-dimensional Styblinski-Tang Function has one global minimum located at $a = -2.903534$, one local minimum located at $d$, and one saddle point $b = 0.156731$ as the boundary separating Valley $a_1$ and Valley $a_2$. For a n-dimensional Styblinski-Tang Function, we initialize parameters $\theta_{t=0} = \frac{1}{\sqrt{k}}(-2.903534, \ldots, -2.903534)$, and set the valley's boundary as $\theta_i < \frac{1}{\sqrt{k}}0.156731$, where $i$ is the dimension index. We record the number of iterations required to escape from the valley to the outside of valley.

**Observation**: we observe the number of iterations from the initialized position to the terminated position. As we are more interested in the number of iterations than "dynamical time" in practice, we use the number of iterations to denote the mean escape time and ignore the time unit $\eta$ in "dynamical time". We repeat experiments 100 times to estimate the escape rate $\Gamma$ and the mean escape time $\tau$. As the escape time is approximately a random variable obeying an exponential distribution, $t \sim Exponential(\Gamma)$, the estimated escape rate can be written as

$$\Gamma = \frac{100 - 2}{\sum_{i=1}^{100} t_i}. \tag{119}$$

The 95% confidence interval of this estimator is

$$\Gamma(1 - \frac{1.96}{\sqrt{100}}) \leq \Gamma \leq \Gamma(1 + \frac{1.96}{\sqrt{100}}). \tag{120}$$

## F  SUPPLEMENTARY EMPIRICAL RESULTS

Some papers (Luo et al., 2019; Chen and Gu, 2018) argued that their proposed Adam variants may generalize as well as SGD. But we found that this argument is contracted with our comparative experimental results, such as Table 1. The main problem may lie in weight decay. SGD with weight decay $\lambda = 0.0001$, a common setting in related papers, is not a good baseline on CIFAR-10 and CIFAR-100, as $\lambda = 0.0005$ often shows better generalization, seen in Figures 9. We also conduct comparative experiments with $\lambda = 0.0001$, seen in Table 2. While some Adam variants under this setting sometimes may compare with SGD due to the lower baseline performance of SGD, Adai and SGD with fair weight decay still show superior test performance.

We display all learning curves of Adai, SGD, and Adam on CIFAR-10 and CIFAR-100 in Figure 10. We further compare convergence of Adai and Adam with the fixed learning rate scheduler for 1000 epochs in Figure 11.

For Language Modeling, we display the results of LSTM in Figure 12.

Finally, we use the measure of the expected minima sharpness proposed by Neyshabur et al. (2017) to compare the sharpness of minima learned by Adai, Momentum, and Adam. The expected minima sharpness is defined as $\mathbb{E}_\zeta[L(\theta^\star + \zeta) - L(\theta^\star)]$, where $\zeta$ is Gaussian noise and $\theta^\star$ is the empirical minimizer learned by a training algorithm. If the loss landscape near $\theta^\star$ is sharp, the weight-perturbed loss $\mathbb{E}_\zeta[L(\theta^\star + \zeta)]$ will be much larger than $L(\theta^\star)$. Figure 13 empirically supports that Adai and Momentum can learn significantly flatter minima than Adam.

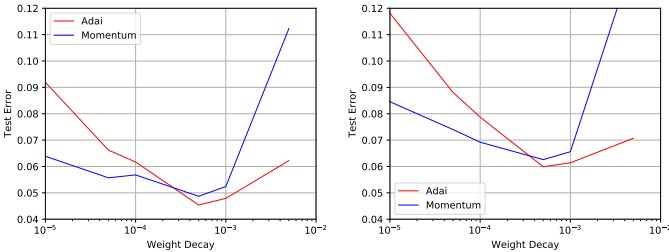

Figure 9: The test errors of ResNet18 on CIFAR-10 and VGG16 on CIFAR-10 under various weight decay. Left: ResNet18. Right: VGG16. The optimal test performance corresponds to $\lambda = 0.0005$. Obviously, Adai has better optimal test performance than SGD with Momentum.

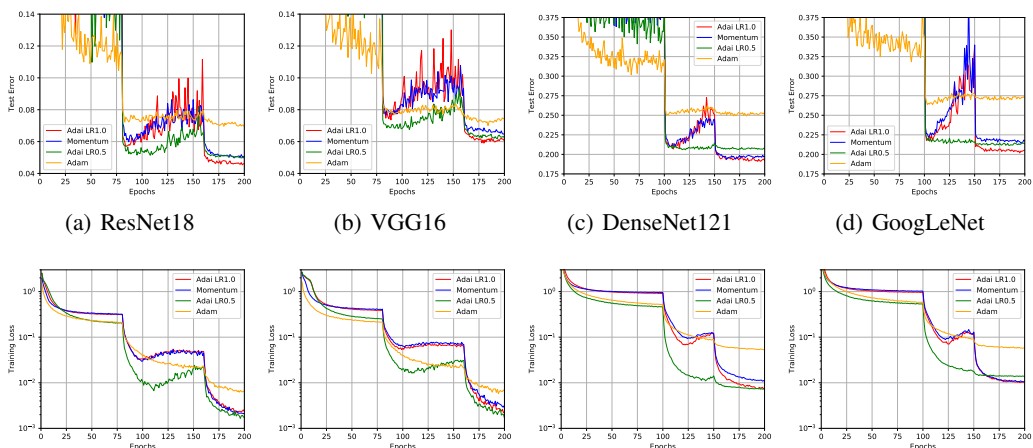

Figure 10: Generalization and Convergence Comparison. Subfigures (a)-(b): ResNet18 and VGG16 on CIFAR-10. Subfigures (c)-(d): DenseNet121 and GoogLeNet on CIFAR-100. Top Row: Test curves. Bottom Row: Training curves. Adai with $\eta = 1$ and $\eta = 0.5$ converge similarly fast to SGD with Momentum and Adam, respectively, and Adai generalizes significantly better than SGD with Momentum and Adam.

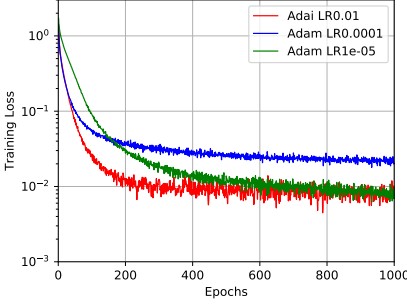

Figure 11: Convergence comparison by training VGG16 on CIFAR-10 for 1000 epochs with the fixed learning rate. When they converge similarly fast, Adai converges in a lower training loss in the end. When they converge in a similarly low training loss, Adai converges faster during training.

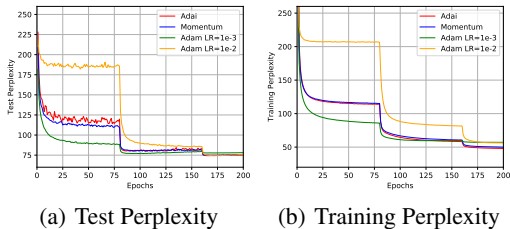

(a) Test Perplexity  (b) Training Perplexity

Figure 12: Language Modeling. The learning curves of Adai, SGD (with Momentum), and Adam for LSTM on Penn TreeBank. The optimal test perplexity of Adai, SGD, and Adam are $74.3$, $74.9$, and $74.3$, respectively. Adai and Adam outperform SGD, while Adai may lead to a lower training loss than Adam and SGD.

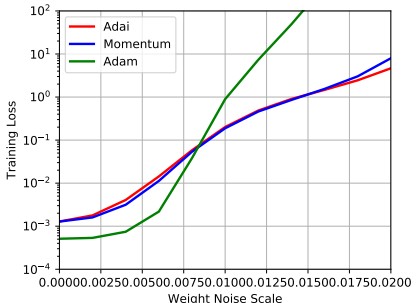

Figure 13: The expected minima sharpness analysis of the weight-perturbed training loss landscape of ResNet18 on CIFAR-10. The weight noise scale is the standard deviation of the injected Gaussian noise. The minima learned by Adai and SGD are more robust to weight noise. Obviously, Adai and Momentum can learn much flatter minima than Adam in terms of the expected minima sharpness.

Table 2: Test performance comparison of optimizers with the weight decay hyperparmeter $\lambda = 0.0001$. In this setting, some Adam variants may compare with SGD mainly because the baseline performance of SGD is lower than the baseline performance in Table 1. The test errors of AdaiW, Adai, and Momentum in middle columns is the original results in Table 1.

| DATASET | MODEL | ADAIW | ADAI | SGD M | SGD M | ADAM | AMSGRAD | ADAMW | ADABOUND | PADAM | YOGI | RADAM |
|---------|-------|-------|------|-------|-------|------|---------|-------|----------|-------|------|-------|
| CIFAR-10 | RESNET18 | $\mathbf{4.59}_{0.16}$ | $4.74_{0.14}$ | $5.01_{0.03}$ | 5.58 | 6.08 | 5.72 | 5.33 | 6.87 | 5.83 | 5.43 | 5.81 |
| | VGG16 | $\mathbf{5.81}_{0.07}$ | $6.00_{0.09}$ | $6.42_{0.02}$ | 6.92 | 7.04 | 6.68 | 6.45 | 7.33 | 6.74 | 6.69 | 6.73 |
| CIFAR-100 | RESNET34 | $21.05_{0.10}$ | $\mathbf{20.79}_{0.22}$ | $21.52_{0.37}$ | 24.92 | 25.56 | 24.74 | 23.61 | 25.67 | 25.39 | 23.72 | 25.65 |
| | DENSENET121 | $\mathbf{19.44}_{0.21}$ | $19.59_{0.38}$ | $19.81_{0.33}$ | 20.98 | 24.39 | 22.80 | 22.23 | 24.23 | 22.26 | 22.40 | 22.40 |
| | GOOGLENET | $\mathbf{20.50}_{0.25}$ | $20.55_{0.32}$ | $21.21_{0.29}$ | 21.89 | 24.60 | 24.05 | 21.71 | 25.03 | 26.69 | 22.56 | 22.35 |

# G   ADAI WITH STABLE/DECOUPLED WEIGHT DECAY

---

**Algorithm 3:** AdaiS/AdaiW

---

$g_t = \nabla \hat{L}(\theta_{t-1})$;

$v_t = \beta_2 v_{t-1} + (1 - \beta_2)g_t^2$;

$\hat{v}_t = \frac{v_t}{1-\beta_2^t}$;

$\bar{v}_t = mean(\hat{v}_t)$;

$\beta_{1,t} = (1 - \frac{\beta_0}{\bar{v}_t}\hat{v}_t).Clip(0, 1 - \epsilon)$;

$m_t = \beta_{1,t}m_{t-1} + (1 - \beta_{1,t})g_t$;

$\hat{m}_t = \frac{m_t}{1-\prod_{z=1}^{t}\beta_{1,z}}$;

$\theta_t = \theta_{t-1} - \eta\hat{m}_t - \lambda\eta\theta_{t-1}$;

---

# H   EXPRESSIONS OF ADAM DYNAMICS

In this section, we discuss why Adam dynamics can also be expressed as Equation (7) similarly to Momentum dynamics.

The derivation that generalizes Momentum dynamics to Adam dynamics is trivial. We only need to replace $\eta$ by the adaptive $\hat{\eta}$ in Equations (4), (5), and (6). We write the updating rule of Adam as

$$\begin{cases} m_t = \beta_1 m_{t-1} + \beta_3 g_t, \\ \theta_{t+1} = \theta_t - \hat{\eta}m_t, \end{cases} \tag{121}$$

where $\beta_3 = 1 - \beta_1$, $\beta_1$ is the hyperparameter, and $\hat{\eta}$ is the adaptive learning rate. For simplicity, it is fine to consider the updating rules as element-wise operation. We can also write the Newtonian motion equations of Adam with the mass $M$ and the damping coefficient $\gamma$ as

$$\begin{cases} r_t = (1 - \gamma dt)r_{t-1} + \frac{F}{M}dt \\ \theta_{t+1} = \theta_t + r_t dt, \end{cases} \tag{122}$$

where $r_t = -m_t$, $F = g_t$, $dt = \hat{\eta}$, $1 - \gamma dt = \beta_1$, and $\frac{dt}{M} = \beta_3 = 1 - \beta_1$. Thus, we obtain the differential-form motion equation of Adam as

$$M\ddot{\theta} = -\gamma M\dot{\theta} + F, \tag{123}$$

where the mass $M = \frac{\hat{\eta}}{1-\beta_1}$ and the damping coefficient $\gamma = \frac{1-\beta_1}{\hat{\eta}}$.

This shows that the analysis can be applied to both Adam dynamics and Momentum dynamics. This is not surprising, because the Newtonian motion equations, including Equations (5) and (6), are universal. The basic updating rules, Equations (4), generally holds for any optimizer that uses Momentum, including Adam. As the terms in Equations (4) can correspond to the terms in Equations (5) one by one, any dynamics governed by Equations (4) can also corresponds to a differential-form Equation (6), where $M$ and $\gamma$ may have different expressions.

We note that Equations (5) in our paper is not contradictory to the SDEs of Adam in Zhou et al. (2020), as long as we let the expressions of $M$ and $\gamma$ follow Adam dynamics. In fact, the updating

rule of $v_t$ of the SDEs in Zhou et al. (2020) can be incorporated into the expressions of $M$ and $\gamma$. Equation (5) in our work, which is a single stochastic differential equation, may be more concise for analyzing optimization dynamics than the SDEs in Zhou et al. (2020).

We also point out that the most important property of using Momentum is to employ the phase-time dynamics for training of deep neural networks. If we let $\beta_1 = 0$ in Adam, Adam will be reduced to RMSprop. Similarly SGD, RMSprop has no the momentum drift effect on saddle-point escaping but has an isotropic diffusion effect. Our theoretical analysis about Momentum and Adam helps us understand how manipulate element-wise momentum and learning rate separately to improve saddle-point escaping as we want.

