# OpenReview forum: "Adaptive Inertia: Disentangling the Effects of Adaptive Learning Rate and Momentum"
_ICLR.cc/2022/Conference — ICLR 2022 Submitted_

### Official Review · Reviewer_EqDN · 2021-10-21

**Correctness:** 2
**Technical Novelty And Significance:** 2
**Empirical Novelty And Significance:** 2
**Recommendation:** 3
**Confidence:** 4

**Main Review:**

Here are some of the questions and concerns:

From my perspective, the theoretical analysis is not rigorous and skips a lot of details. For example, in Section 4 (Page 5), the paper claims that the continuous-time dynamic of Adam can be written as equation (6). However, a detail derivation to support the claim and the proof of showing that the SDE is a valid one for Adam are missing in the paper. Indeed, it can be seen from Page 5, where the authors said they are analyzing an "idealized" Adam. It is not clear why this is an idealized one and the writing is like hiding something under the rug. We can see that some statements about Adam are made, but these conclusions are drawn from a SDE which does not really correspond to Adam. On the other hand, the SDE of Adam does exist in the literature, e.g. https://arxiv.org/pdf/2010.05627.pdf. If this paper really wants to claim/argue something about Adam, then a careful analysis regarding the discretization error between the solution of the proposed SDE and the discrete-time Adam should be provided in the paper.

For another example, on page 3, it states QUOTE the diffusion matrix D is independent of $\theta$ near critical points UNQUOTE. But the proof is not provided. How is it independent? It looks like some approximations were used there. The paper should prove the "independence".

Another concern is about (8). The paper should provide a detailed derivation of showing that (8) is really about how the distribution evolves when the underlying dynamic is (7). Currently the equation (8) is like jumping out of nowhere. How does Assumption 1 help to show (8)?

Also, some places in the paper are not clear:

(a) (Second paragraph on Page 1) QUOTE all previous works have not touched the saddle-point escaping property of the dynamics UNQUOTE Apparently there are quite a few works regarding saddle-point escaping by SGD, SGD with momentum, and Adam. The authors might want to explain what they meant here.

(b) There are two approximations on (3). But it would be more helpful to explain how the approximations are made in detail. There are some descriptions below (3) but are not very clear.

(c) (Last paragraph in Section 3) QUOTE The momentum does not affect flat minima selection in terms of the escape time UNQUOTE This is another confusing statement. What does "momentum does not affect flat minima selection" mean? Does it mean momentum and SGD without momentum converge to the same point? What is the definition of "flat minima"?

(d) (Second to last paragraph in Section 4) QUOTE Adam has $\log(\tau)=O(H_{ae}^{-1/2})$ ... SGD and SGD+momentum both have $log(\tau) = O(H_{ae}^{-1})$ UNQUOTE It seems that the conclusion right below this sentence would be reversed if $|H_{ae}|>1$. Something wrong?

(e) (Theorem 1 and 2) The authors show some guarantees about the variance at time $t$ when the iterate of the algorithm follows the Gaussian distribution. But does a higher variance of the Gaussian distribution imply a faster saddle point escape? The authors might want to add some discussions about the connection to the notion of saddle point escape in the literature (e.g  Jin et al. 2017).

(f) After reading the paper, I am still not sure how the effect of the learning rate and momentum was "disentangled" in the analysis. I see some analysis about the behavior of SGD, ADAM, SGD+momentum at critical points.  It would be more helpful if the authors can explain why the effect of the learning rate and momentum can be isolated.



**Summary Of The Paper:**

This paper studies the behaviors of some algorithms when the iterate is at a critical point via SDEs. A variant of Adam is given in the end. The paper draws some conclusions about adaptive learning rate and momentum, claiming that

- QUOTE momentum matters little to escaping sharp minima UNQUOTE
- QUOTE adaptive learning rate is not good at selecting flat minima UNQUOTE


**Summary Of The Review:**

The presentation and statements are confusing in my opinion. Some steps in the analysis are not transparent.

---

> ### Author Response · Authors · 2021-11-17
> **Responses (1) to Reviewer EqDN**
>
> We appreciate the reviewer for the hard work and helpful comments.
>
> The main concerns have been duly addressed below.
>
>
> Q1: The paper claims that the continuous-time dynamic of Adam can be written as equation (6). A detail derivation to support the claim and the proof of showing that the SDE is a valid one for Adam are missing in the paper.
>
> A1: The derivation that generalizes Momentum dynamics to Adam dynamics is trivial. Briefly speaking, we only need to replace $\eta$ by the adaptive $\hat{\eta}$ in Equations (4) – (6). Then the analysis can be applied to both Adam and Momentum. This is not surprising, because the Newtonian motion equations, including Equations (5) and (6), are universal. We will present the derivation in Appendix in the revised version.
>
> We also note that the basic update rules, Eqs (4), generally holds for any optimizer that uses Momentum, including Adam. Eq (6) and Eqs (5) are the two forms of the motion equation in Newtonian Mechanics. As the terms in Eqs (4) can correspond to the terms in Eqs (5) one by one, any dynamics governed by Eqs (4) can also corresponds to a differential-equation form Eq (6), where $M$ and $\gamma$ may have different expressions. As both Momentum and Adam can be written in Eqs (4), Adam can be written in Eq (6).
>
>
> Q2: Why does the SDE of Adam looks different from the form in [1].
>
> A2: Eq (6) in our paper is not contradictory to the SDE of Adam in [1], as long as we specify the expressions of $M$ and $\gamma$ as Adam’s. In fact, the updating rule of $v_{t}$ in [1] can be incorporated into the expressions of $M$ and $\gamma$. However, our Eq (6), which mainly includes a single differential equation, may be more concise for analyzing optimization dynamics. Thanks for the comment. We will point out this relation in the revised to make the points more clear.
>
>
> Q3: How about the discretization error between the solution of the proposed SDE and the discrete-time Adam?
>
> A3: We agree that the discretization error between the solution of the proposed SDE and the discrete-time Adam is a very important topic. The discretization error analysis for Adam is absent in both [1] and our work. However, the discretization error analysis is beyond the scope of both [1] and our work. For example, one reason why we need to analyze the discretization error is to analyze the weak approximation/convergence in SDE, i.e., when averaging a particular function over a sample path obtained in SDE. However, this paper is not intended to analyze such things. The reason why [1] and our paper does not analyze the discretization error is also because it is not the purpose of the problem to be solved in practical use of Adam.
>
>
> Q4: Why is the diffusion matrix $D$ is independent of $\theta$ near critical points?
>
> A4: As we mentioned on the top of Page 3, it is a direct result of Eq (3) and Assumption 1. Eq (3) demonstrates that $D$ is proportional to the Hessian $H$ near a critical point $\theta^{\star}$, while the Hessian is equal to $H(\theta^{\star})$ according to Assumption 1. This can also be empirically verified by our experimental rults in Figure 6.
>
>
> Q5: Page 5, the authors said they are analyzing an "idealized" Adam. It is not clear why this is an idealized one.
>
> A5: We argued that the “idealized” Adam in our paper has a clear meaning that the diagonal $v$ can be regarded as a diagonal approximation of the full matrix $\mathbb{E}[g_{t}g_{t}^{\top}]$. This diagonal approximation $v=\mathbb{E}[g_{t}g_{t}^{\top}]$ is common and also applied by previous work [2], which also theoretically studied how Adam escapes saddle points.
>
> Q6: Why does Eq (8) describe the probability density distribution? How does Assumption 1 help to show Eq (8)?
>
> A6: The Eq (8) is the famous Fokker-Planck Equation (in phase space), which describes the probability density distribution given by Langevin Dynamics. This was a well-known result even decades ago [3-4]. Under Assumption 1, the diffusion matrix $D$ is constant near critical points, which is the same as statistical physics. Thus, we may directly use Eq(8) in statistical physics.
>
>
> We borrowed the well-known Eq (8) from physics to make progress in optimization dynamics of deep learning. Note that Eq (8) is not specialized for optimization dynamics and is not our contribution. Our contribution is obtaining novel results by first applying Eq (8) to deep learning. For quickly checking the correctness of Eq (8), please have a look at Eq (57) of [5] (https://www.thphys.uni-heidelberg.de/~wolschin/statsem20_7s.pdf), which used exactly same terms as Eq(8) in our paper. We will outline this relation and make our contribution more clear in the revised version.
>
>
> Please see Responses (2) for more discussions.

---

> ### Author Response · Authors · 2021-11-17
> **Responses (2) to Reviewer EqDN**
>
>
> Q7:  Why does “all previous works have not touched the saddle-point escaping property of the dynamics” mean?
>
> A7: We mean previous works using diffusion theory/SDE did not touch the saddle-point escaping property of the dynamics, while we cited some works (Dauphin et al., 2014; Staib et al., 2019; Jin et al., 2017; Reddi et al., 2018) on the saddle-point escaping property using other tools. Our analysis using diffusion theory provides novel insights beyond existing papers.
>
>
> Q8: Why do two approximations mean in Eq (3)?
>
> A8: Eq (3) is a classical conclusion. It was introduced to SGD dynamics by [6] years ago. The first approximation means “the gradient variance dominates the gradient expectation near critical points”(as the line above Eq(3) indicates). The second approximation means that the observed Fisher Information matrix is approximately equal to the Hessian near critical points. This supported by the listed references.
>
> Q9: What does “the momentum does not affect flat minima selection in terms of the escape time” mean?
>
> A9: Our point is clear in the last paragraph of Section 3. We compare how the mean escape time of Momentum/SGD depends on the Hessian, which can be a measure of minima flatness. “If we properly choose the learning rate, i.e., $\eta_{\mathrm{Momentum}} = \frac{1-\beta_1}{\beta_3} \eta_{\mathrm{SGD}}$,” Momentum and SGD can escape sharp minima similarly fast. More specifically, the time-Hessian relation $\log(\tau) = \mathcal{O}\left(H_{ae} \right) $ holds for both SGD and Momentum, but does not hold for Adam.
>
>
> Q10: Adam has $\log(\tau) = \mathcal{O}\left(H_{ae}^{\frac{1}{2}} \right) $, while SGD/Momentum has $\log(\tau) = \mathcal{O}\left(H_{ae} \right) $. Will the conclusion be reversed if  $\|H_{ae}\| > 1$, given that?
>
> A10: No, the conclusion will not be reversed. The key point here is not the magnitude of the mean escape time but how the magnitude depends on $H_{ae}$. The key point is that the mean escape time of Adam is significantly less dependent on the minima sharpness than both Momentum and Adai.
>
> For example, we use $k$ to control the Hessian in the empirical analysis of the mean escape time (on Page 9). Assume that we use $k=1$ as the baseline mean escape time. If we increase $k$ to make minima sharper, the mean escape time of SGD/Momentum will always decrease faster than Adam compared to the baseline mean escape time. If we decrease $k$ to make minima flatter, the mean escape time of SGD/Momentum will always increase faster than Adam compared to the baseline mean escape time.
>
>
> Q11: Does a higher variance of the Gaussian distribution imply a faster saddle point escape?
>
> A11: It is true for vanilla SGD dynamics, but does not hold for momentum-based dynamics. We actually use the mean squared displacement to measure saddle-point escaping. When the probability density function is zero-centered Gaussian (such as SGD dynamics), the mean squared displacement is exactly the variance of the Gaussian distribution. The mean squared displacement is indeed different from the measure in the conventional line [7], but provides us novel insights. We believe that introducing this important measure from statistical physics into optimization dynamics can help the community better understand optimization dynamics from a new perspective. Thanks for your suggestion. We will make this point more clear in the revised version.
>
> Please see Responses (3) for more discussions.

---

> ### Author Response · Authors · 2021-11-17
> **Responses (3) to Reviewer EqDN**
>
> Q12: It would be more helpful if the authors can explain why the effect of the learning rate and momentum can be isolated.
>
> A12: It means we can understand how changing learning rate or momentum may change optimization dynamics near critical points and minima. For example, Momentum provides the momentum drift term for escaping saddle points. Thus, we may manipulate momentum to improve saddle-point escaping. Because, in the theorems and propositions of Section 3-4, we mathematically separated how Adaptive Learning Rate and Momentum affect saddle-point escaping and minima selection. This is the prerequisite of designing Adai.
>
>
> References:
>
> [1]Zhou, P., Feng, J., Ma, C., Xiong, C., & Hoi, S. C. H. (2020). Towards Theoretically Understanding Why Sgd Generalizes Better Than Adam in Deep Learning. Advances in Neural Information Processing Systems, 33.
>
> [2] Staib, M., Reddi, S., Kale, S., Kumar, S., & Sra, S. (2019, May). Escaping saddle points with adaptive gradient methods. In International Conference on Machine Learning (pp. 5956-5965). PMLR.
>
> [3] Risken, H. (1996). Fokker-planck equation. In The Fokker-Planck Equation (pp. 63-95). Springer, Berlin, Heidelberg.
>
> [4] Risken, H., & Eberly, J. H. (1985). The fokker-planck equation, methods of solution and applications. Journal of the Optical Society of America B Optical Physics, 2(3), 508.
>
> [5] Radpay, P. (2020). Langevin Equation and Fokker-Planck Equation.
>
> [6] Zhu, Z., Wu, J., Yu, B., Wu, L., & Ma, J. (2019, May). The Anisotropic Noise in Stochastic Gradient Descent: Its Behavior of Escaping from Sharp Minima and Regularization Effects. In International Conference on Machine Learning (pp. 7654-7663). PMLR.
>
> [7] Jin, C., Ge, R., Netrapalli, P., Kakade, S. M., and Jordan, M. I. (2017). How to escape saddle points efficiently. In International Conference on Machine Learning, pages 1724–1732. PMLR.

---

### Official Review · Reviewer_A3Wq · 2021-11-02

**Correctness:** 3
**Technical Novelty And Significance:** 3
**Empirical Novelty And Significance:** 3
**Recommendation:** 6
**Confidence:** 3

**Main Review:**

This paper is generally well-written. The diffusion theoretical analysis does provide some insight on the empirical performance of momentum SGD and Adam. The authors also put in efforts to conduct numerical verifications to their theoretical statements, which is highly appreciated. However, I think that this work does not completed ''disentangle'' the effects of adaptive learning rate and momentum since the work analyzes Adam, which fuses these two algorithmic components. It would be better to discuss the effect of each component in Adam separately (probably by setting some parameters to zero).

The authors then propose Adai, which achieves approximately Hessian-independent momentum drift without damaging flat minima selection (BTW, the proof of Proposition 3 is missing, if it is a direct consequence of Theorem 2, it is better to mention it somehow). The construction of Adai is interesting, and its effectiveness is justified by the empirical experiments. However, it seems to me that this contribution is a bit disconnected to the main story as Adai does not use adaptive learning rate. Some revision (probably changing the title?) might be good to make the story clearer and more fluent.

Typos:
- Missing reference on page 7, "Note that an existing “adaptive momentum” method (?) "
- The last sentence "better than popular Adam and SGD".

**Summary Of The Paper:**

This work analyzes the dynamics of momentum SGD and Adam on escaping saddle points and sharp minima, which is based on the diffusion theoretical framework proposed in (Xie et al. 2021b). The authors prove that momentum provides a drift effect around saddle points and does not affect flat minima selection (for SGD), and while Adam escapes saddle points efficiently, it does not favor flat minima as well as SGD. The analysis explains some empirical observation of SGDM and Adam. Motivated by the analysis, the authors propose adaptive inertia (Adai) method, which can approximately achieve Hessian-independent momentum drift (escapes saddle points fast) and favors flat minima as well as (momentum) SGD.

**Summary Of The Review:**

This work provides some new theoretical insights for momentum SGD and Adam, which are interesting and important. The authors then propose adaptive inertia based on the insights, which shows good performance. Some revision is needed to make the story clearer (see main review).

---

> ### Author Response · Authors · 2021-11-17
> **Responses to Reviewer A3Wq**
>
> We appreciate the reviewer for the helpful comments and the kind support to our work.
>
> We have addressed your main concerns as follows.
>
> Q1: This work does not completed ''disentangle'' the effects of adaptive learning rate and momentum since the work analyzes Adam, which fuses these two algorithmic components. It would be better to discuss the effect of each component in Adam separately (probably by setting some parameters to zero).
>
> A1: Thanks for the suggestion. We would like to add more discussions on the effect of each component separately in the revised version. It may help readers intuitively understand each component better.
>
> Q2: Is the proof of Proposition 3 missing? If it is a direct consequence of Theorem 2, it is better to mention it somehow.
>
> A2: Yes, Proposition 3 is a direct consequence of Theorem 2. We will note this point in the revised version. Thanks for carefully reading our theoretical analysis.
>
> Q3: The contribution of Adai is a bit disconnected to the main story as Adai does not use adaptive learning rate. Some revision (probably changing the title?) might be good to make the story clearer and more fluent.
>
> A3: Thanks for the helpful comments. One may design an optimizer like Adai only if we know the theoretical analysis of separating Adaptive Learning Rate and Momentum. As we found that Adaptive Learning Rate is not good at selecting flat minima, using adaptive momentum inertia becomes a nice choice and is beyond conventional adaptive gradient methods. We will try to make the story more fluent.
>
> Thanks for pointing out the typos. We will correct them in the revised version.
>
> We sincerely thanks again for your constructive comments and suggestions.

---

### Official Review · Reviewer_MYxM · 2021-11-02

**Correctness:** 4
**Technical Novelty And Significance:** 3
**Empirical Novelty And Significance:** 3
**Recommendation:** 8
**Confidence:** 4

**Main Review:**

The theoretical analysis of the Adam optimizer is based on the SGD diffusion theory, and the results confirms and explains the observation that Adam can sometimes converge faster but generalize worse than SGDM. The proposed Adai optimizer is theoretically sound, and demonstrates slightly better generalization performance than SGDM (and significantly better than Adam) on image classification tasks.

Despite estimating the moments in a similar way as Adam, the proposed Adai optimizer seems more akin to SGDM, with the only difference being its adaptive momentum; and it doesn't use adaptive learning rates, which is a main feature of Adam. Moreover, as shown in Figs. 1 to 3, and 10, the training curves of Adai very much resemble those of SGDM. Therefore, to further improve this work, more comparisons should be made between Adai and SGDM rather than between Adai and Adam. In particular, it would be interesting to see if the performance gap between Adai and SGDM results from faster convergence (as suggested by the theory), and therefore a convergence comparison between Adai and SGDM as the one conducted between Adai and Adam (Fig. 11) should be helpful.

**Summary Of The Paper:**

This paper disentangles the effects of adaptive learning rate and momentum in Adam learning dynamics, and proves that adaptive learning rate is good at escaping saddle points but not good at selecting ﬂat minima, while momentum helps escape saddle point and matters little to escaping sharp minima. Based on the analysis, the authors propose a novel optimizer, Adai. Compared to SGDM, Adai parameter-wisely adapts the momentum hyperparameter to the (approximated) Hessians of saddle points, and is proved to fast escape saddle points and sharp minima.

**Summary Of The Review:**

This paper provides new insights into the performance of Adam, and proposes a novel optimizer that both converges fast and generalizes well. Further improvements can be made by comparing the proposed method to SGDM more thoroughly.

---

> ### Author Response · Authors · 2021-11-17
> **Responses to Reviewer MYxM**
>
> We appreciate the reviewer for the helpful comments and the kind support to our work.
>
> We have addressed the main concerns as follows.
>
>
> Q1:  The proposed Adai optimizer seems more akin to SGDM, with the only difference being its adaptive momentum. Therefore, to further improve this work, more comparisons should be made between Adai and SGDM rather than between Adai and Adam.
>
> A1: Thanks for the suggestion. We will try to make comparisons between Adai and SGDM, while Adai has showed significant advantages in empirical analysis, such as Figures 2 and 4.
>
>
> Q2: In particular, it would be interesting to see if the performance gap between Adai and SGDM results from faster convergence (as suggested by the theory).
>
> A2: In the experiments on CIFA-10 and ImageNet, the training curves of Figure 2 and Figure 3 both suggest that Adai usually shows faster convergence and lower training losses than SGDM. This can help understand the performance improvement of Adai over SGDM. We also think that it will be helpful to further empirically verify if Adai can escape saddle points faster than SGDM in deep learning. However, given various hyperparameters and many saddle points of deep loss landscape, it is hard to directly compare saddle-point escaping alone of Adai and Momentum in the real-world experiments.
>
> Thanks a lot for the kind support and constructive comments again.

---

### Official Review · Reviewer_1sdV · 2021-11-03

**Correctness:** 2
**Technical Novelty And Significance:** 3
**Empirical Novelty And Significance:** 3
**Recommendation:** 5
**Confidence:** 4

**Main Review:**

I believe that the main ideas of the paper are interesting. However I find that the presentation of this work is not very clear and somehow confusing. In particular, the structure of the paper and the results presented in sections 2 and 3 are difficult to absorb. See my comments below.

I understand the motivation of the authors and what they tried to communicate but i find that there is no satisfactory explanation of the results presented in sections 2 and 3. The authors assumed that the reader is familiar with the closely related recent work on the SGD diffusion theory and do not provide enough details on the framework. For example they use terminology like "Fokker-Planck equation",  "divergence operator", and "diffusion matrix" that are not really standard in the area of adaptive methods. In addition, Assumption 1 on the second order Taylor approximation near critical points, is given without providing some interesting problems where it is satisfied.

Also in section 3, Assumptions 2 and 3 are used without further explanation of what exactly they mean. The authors provide a few details in the appendix on what is the quasi-equilibrium approximation and low temperature approximation but this is not sufficient.  How these assumptions are related to standard concepts in the area? What are the mathematical expressions of these assumptions? how are related to stochastic gradients and the noise?

Also i find it a bit surprising that there is no formal presentation of the problem that we are interested to solve and the assumptions that one requires to be able to prove convergence. A statement of the minimization (or maximation) problem under study with the main assumptions is missing from the paper.

One of the most important contributions of the paper is the analysis of the new algorithm, Adaptive Inertia Optimization (Adai) proposed in Section 5. However if one focuses on Theorem 4 which provides the convergence guarantees of the Adai it is clear that the analysis hold under very strong conditions / assumptions. For example the authors assumed both bounded variance, and bounded gradient of the objective function, which rarely hold in practical scenarios. Note that these conditions have already been proved to contradict special classes of non-convex problems like functions satisfying the Polyak-Lojasiewicz condition (the combination of these assumptions lead to an empty set of problems). Thus the theorem cannot hold for all non-convex smooth problems.

**Summary Of The Paper:**

The paper focuses on the understanding of the effects of adaptive learning rate and momentum. In particular it proves that the adaptive learning rate can escape saddle points efficiently and cannot select flat minima as SGD does. It also shows that momentum helps the training process by passing through saddle points and without affecting the minima selection.The paper also proposes a new adaptive algorithm, named Adai (Algorithm 2), which uses parameter-wise adaptive intertia to accelerate the training and finds flat minima as well as SGD.  Finally, the paper provides extensive numerical testing showing the benefits of Adai.

**Summary Of The Review:**

As i mentioned in the main review, I believe that the main ideas of the paper are interesting. However I find that the presentation of this work is not very clear and somehow confusing. In particular, the structure of the paper and the results presented in sections 2 and 3 are difficult to absorb. See my comments below.

---

> ### Author Response · Authors · 2021-11-17
> **Responses (1) to Reviewer 1sdV**
>
> We appreciate the reviewer for the helpful comments and your interests in our work.
>
> Your concerns are mainly about the assumptions / conditions used by our work.
>
> We argue that the assumptions / conditions in our paper are indeed common and mild in related papers.
>
> We would like to supply related references and discussions for better supporting our work.
>
>
> Q1:  Why did you use terminology like "Fokker-Planck equation", "divergence operator", and "diffusion matrix" that are not really standard in the area of adaptive methods?
>
> A1: We kindly argued that the concepts are very common in basic math, stochastic process, or the line of Langevin-Dynamics analysis of optimization dynamics. For example, "divergence operator" is a basic mathematic operator and common in the college-level Calculus course. The work of SGD diffusion theory actually did not define novel concepts. The "Fokker-Planck Equation" describes the probability density function (posterior) produced by Langevin Dynamics. It is a basic equation in a lot of related papers on Langevin-Dynamics-based analysis of optimization dynamics and stochastic process, including [1-5]. The terminology "diffusion matrix" does not harm the clarity because it is just a common name of $D$ often used in related papers. We will try to clarify them better in the revised version.
>
>
> Q2: Assumption 1 on the second order Taylor approximation near critical points, is given without providing some interesting problems where it is satisfied.
>
> A2: Assumption 1 means that our theoretical analysis focused on the optimization behaviors near critical points. We argue that the second-order Taylor approximation is actually very widely used in related papers [7-10], as we mentioned in the main text. We agree that it will be possible to explore the more refined high-order theory in future, while the high-order terms are usually much smaller than the second-order term.
>
>
> Q3: Assumptions 2 and 3 are used without further explanation of what exactly they mean.
>
> A3: Our work follows the existing diffusion theory for analyzing minima escaping processes, where Assumptions 2 and 3 are common in both deep learning and statistical physics literatures (Please see Page 4). We actually presented more discussions on these two classical assumptions in Appendix B due to the page limit of the main text, while the existing diffusion theoretical framework is not our contribution. Appendix B helps clarify the exact meaning of the two assumptions. The empirical results also support that they are mild in our studied problems.
>
>
>
> Q4: Theorem 4 (the convergence guarantee of the Adai) holds under very strong conditions / assumptions. For example the authors assumed both bounded variance, and bounded gradient of the objective function, which rarely hold in practical scenarios. Thus, the theorem cannot hold for all non-convex smooth problems.
>
> A4: We admit that Theorem 4 indeed cannot hold for all non-convex problems. We argue that Adai has similar convergence guarantee to other momentum-based stochastic optimization methods, which recent work [11] studied comprehensively. Bounded variance and bounded gradients are still common in the convergence analysis of momentum-based stochastic optimization methods [11-13]. Adai, as an adaptive momentum method, also adopts the two conditions. However, we agree that it may be possible to further relax the convergence assumptions of Adai and other momentum-based optimizers, while this is beyond the scope of our main contributions.
>
> Please see Responses (2) for more discussions.

---

> > ### Comment · Reviewer_1sdV · 2021-11-29
> > **Review Update**
> >
> > Thank you to the authors for providing further clarification to the raised points.  I have read the other reviews, the rebuttal and browsed through the paper again.
> >
> > I still believe that the presentation of sections 2 and 3 is not clear enough and that several theoretical statements are not rigorous.
> > For these reasons, I believe that my original score was fair.
> > I also encourage the authors to incorporate the reviewers' suggestions in the future updates of their work.
> >
> > Let me provide more details on the above statement:
> >
> > In terms of presentation, the authors assume that some recent results on "SGD diffusion theory" are common knowledge. However, to the best of my knowledge, using a diffusion theory to explain the behavior of a novel adaptive algorithm is a new concept and it should be presented in a more rigorous way. I understand that some of the concepts are very common in basic math, stochastic process, or the line of Langevin-Dynamics analysis of optimization dynamics, but this does not mean that people interested in adaptive optimization algorithms are familiar with them. The paper essentially proposes a new adaptive optimization algorithm. As such, I believe that it should have a clearer connection between the two literatures (diffusion theory and adaptive optimization literature).
> >
> > Nevertheless, the presentation is not the reason I kept my score the same. I understand that this can be improved in the camera-ready version. My main concern is the assumptions used to prove convergence rates. The assumptions are very strong, and in several practical scenarios, they are not really satisfied. That is, they hold for an empty set of functions.
> > For example, if it turns out that the problem has all local minima to be global (let's say that it satisfies the Polyak-Lojasiewicz condition), then the results of the proposed Theorems hold for an empty set of functions. In their response, the authors admit that Theorem 4 cannot hold for all non-convex problems, but they did not propose a fix for this issue. They claim that bounded variance and bounded gradients are still common in the convergence analysis of momentum-based stochastic optimization methods. I'm afraid I have to disagree with this statement; many recent papers remove such strong assumptions and show better convergence guarantees. See, for example, references [1] and [2] below.
> >
> > Also, Assumptions 2 and 3 are still not rigorous enough. Even if the authors put some details about them in the appendix, to the best of my knowledge, assumptions named "Quasi-Equilibrium Approximation" and "Low Temperature Approximation" are not standard in the analysis of adaptive methods. In my opinion, there is still no clear explanation in the paper on what these conditions mean mathematically and how they are connected to the rest of the assumptions.
> >
> > References:
> >
> > [1] Liu, Yanli, Yuan Gao, and Wotao Yin. "An Improved Analysis of Stochastic Gradient Descent with Momentum." Advances in Neural Information Processing Systems 33 (2020).
> >
> > [2] Sebbouh, Othmane, Robert M. Gower, and Aaron Defazio. "Almost sure convergence rates for stochastic gradient descent and stochastic heavy ball." In Conference on Learning Theory, pp. 3935-3971. PMLR, 2021.

---

> > > ### Author Response · Authors · 2021-12-03
> > > **Responses to Update Review of Reviewer 1sdV**
> > >
> > > We appreciate Reviewer 1sdV’s updated review.
> > >
> > > We kindly argue that your main concern may be addressed.
> > >
> > > We would like to present two more responses as follows.
> > >
> > > Q1: The assumptions (bounded gradients and bounded variance) for convergence analysis of Adai is strong. Recent papers that focus on convergence analysis [1,2] removed such strong assumptions and show better convergence guarantees.
> > >
> > > A1: Thanks for the suggestion and the references. We agree that, relaxing the bounded gradients or bounded variance assumption like [1,2] may provide better convergence guarantees. However, it does not necessarily mean that all convergence analysis that used bounded gradients and bounded variance in other recent papers (See Reponses (1)) is meaningless. Relaxed convergence guarantees may exist because providing the improved or optimal convergence guarantees may be not the main contribution or purpose of these papers unlike [1].
> > >
> > > Moreover, we think it can be trivial to generalize the convergence analysis in [1] for Adai, because Adai is a special case of Multistage SGDM (See Algorithm 1 in [1]). Thus, Adai may have similar convergence guarantees to Multistage SGDM under the milder assumptions. Note that Multistage SGDM has stagewise momentum hyperparameters and the convergence analysis is dimensionality-free. Adai is actually a kind of Multistage SGDM that uses only one iteration per stage. The property of Adai that the momentum ``hyperparameters’’ are updated algorithmically still makes Adai a kind of SGDM where the momentum hyperparameters are arbitrarily chosen from $[0, 1 )$.
> > >
> > >
> > > Q2: Assumptions 2 and 3 are still not rigorous enough. Even if the authors put some details about them in the appendix, to the best of my knowledge, assumptions named "Quasi-Equilibrium Approximation" and "Low Temperature Approximation" are not standard in the analysis of adaptive methods. In my opinion, there is still no clear explanation in the paper on what these conditions mean mathematically and how they are connected to the rest of the assumptions.
> > >
> > >
> > > A2: Assumptions 2 and 3 and are common in stochastic optimization dynamics of deep learning, and adaptive optimization dynamics is a specialized case of stochastic optimization dynamics which uses various learning rates. We introduced Assumptions 2 and 3 in details even beyond most related papers [4-5]. We argue that our discussion in Page 4-5 and Appendix B may explain and justify Assumptions 2 and 3 as previous papers [3-5].
> > >
> > > We also kindly argue that we discussed the mathematical meaning of Assumptions 2 and 3 in Page 4-5 and Appendix B.
> > >
> > > Assumption 2 means that $\frac{\partial P(\theta, t)}{\partial t} \approx  0 $ holds near minima, but not necessarily holds near saddle points. Quasi-Equilibrium Assumption is actually weaker but more useful than the conventional stationary assumption for deep learning, which requires $\frac{\partial P(\theta, t)}{\partial t} \approx  0 $ everywhere.
> > >
> > > Assumption 3 is mild and justified when $\frac{\eta}{B}$ is small. Under Low Temperature Assumption, the probability densities will concentrate around minima. Numerically, the 6-sigma rule may often provide good approximation for a Gaussian distribution. We even provided an intuitive explanation about Low Temperature Assumption in the last paragraph of Appendix. Numerically, the approximation is mild when the condition $\frac{\Delta L}{H} > 2.3 \times 10^{-4}$ holds well in optimization dynamics. Empirically, [3] reported that the escape processes in the wide range of iterations (50 to 100,000 iterations) can be modeled as a Kramers Escape Problem very well.
> > >
> > > References:
> > > [1] Liu, Yanli, Yuan Gao, and Wotao Yin. "An Improved Analysis of Stochastic Gradient Descent with Momentum." Advances in Neural Information Processing Systems 33 (2020).
> > >
> > > [2] Sebbouh, Othmane, Robert M. Gower, and Aaron Defazio. "Almost sure convergence rates for stochastic gradient descent and stochastic heavy ball." In Conference on Learning Theory, pp. 3935-3971. PMLR, 2021.
> > >
> > > [3] Xie, Z., Sato, I., & Sugiyama, M. (2020, September). A Diffusion Theory For Deep Learning Dynamics: Stochastic Gradient Descent Exponentially Favors Flat Minima. In International Conference on Learning Representations.
> > >
> > > [4] Zhou, P., Feng, J., Ma, C., Xiong, C., & Hoi, S. C. H. (2020). Towards Theoretically Understanding Why Sgd Generalizes Better Than Adam in Deep Learning. Advances in Neural Information Processing Systems, 33.
> > >
> > > [5] Jastrzębski, S., Kenton, Z., Arpit, D., Ballas, N., Fischer, A., Bengio, Y., & Storkey, A. (2017). Three factors influencing minima in sgd. arXiv preprint arXiv:1711.04623.

---

> ### Author Response · Authors · 2021-11-17
> **Responses (2) to Reviewer 1sdV**
>
>
> Q5: A statement of the minimization (or maximation) problem under study with the main assumptions is missing from the paper.
>
> A5: The optimization problem targeted in this paper is a general one as $ \min_{\theta} L(\theta) $. In this type of study, the optimization problem to be solved is general or in common, and the difference in the update formula is more important. Thus, the existing study typically describes the equation for parameter update, omitting the optimization problem. For example, Adam[14] and AdamW[15] omitted the formulation. Therefore, we followed the papers and omitted this statement due to a lack of space. We would like to add the formulation in the revised version.
>
>
>
> References:
>
> [1] Sato, I., & Nakagawa, H. (2014, June). Approximation analysis of stochastic gradient Langevin dynamics by using Fokker-Planck equation and Ito process. In International Conference on Machine Learning (pp. 982-990). PMLR.
>
> [2] Zhang, C., Liao, Q., Rakhlin, A., Miranda, B., Golowich, N., & Poggio, T. (2018). Theory of deep learning IIb: Optimization properties of SGD. arXiv preprint arXiv:1801.02254.
>
> [3] Mou, W., Wang, L., Zhai, X., & Zheng, K. (2018, July). Generalization bounds of sgld for non-convex learning: Two theoretical viewpoints. In Conference on Learning Theory (pp. 605-638). PMLR.
>
> [4] Gao, Y., Jiao, Y., Wang, Y., Wang, Y., Yang, C., & Zhang, S. (2019, May). Deep generative learning via variational gradient flow. In International Conference on Machine Learning (pp. 2093-2101). PMLR.
>
> [5] Xie, Z., Sato, I., & Sugiyama, M. (2020, September). A Diffusion Theory For Deep Learning Dynamics: Stochastic Gradient Descent Exponentially Favors Flat Minima. In International Conference on Learning Representations.
>
> [6] Hoffman, M., & Ma, Y. (2020, November). Black-Box Variational Inference as a Parametric Approximation to Langevin Dynamics. In International Conference on Machine Learning (pp. 4324-4341). PMLR.
>
> [7] Zhang, G., Li, L., Nado, Z., Martens, J., Sachdeva, S., Dahl, G., ... & Grosse, R. B. (2019). Which algorithmic choices matter at which batch sizes? insights from a noisy quadratic model. Advances in neural information processing systems, 32, 8196-8207.
>
> [8] Li, Q., Tai, C., & Weinan, E. (2017, July). Stochastic modified equations and adaptive stochastic gradient algorithms. In International Conference on Machine Learning (pp. 2101-2110). PMLR.
>
> [9] Mandt, S., Hoffman, M. D., & Blei, D. M. (2017). Stochastic Gradient Descent as Approximate Bayesian Inference. Journal of Machine Learning Research, 18, 1-35.
>
> [10] Zhou, P., Feng, J., Ma, C., Xiong, C., & Hoi, S. C. H. (2020). Towards Theoretically Understanding Why Sgd Generalizes Better Than Adam in Deep Learning. Advances in Neural Information Processing Systems, 33.
>
> [11] Yan, Y., Yang, T., Li, Z., Lin, Q., & Yang, Y. (2018, January). A unified analysis of stochastic momentum methods for deep learning. In IJCAI International Joint Conference on Artificial Intelligence.
>
> [12] Xie, Z., Yuan, L., Zhu, Z., and Sugiyama, M. (2021). Positive-negative momentum: Manipulating stochastic gradient noise to improve generalization. In International Conference on Machine Learning.
>
> [13] Ghadimi, S., & Lan, G. (2013). Stochastic first-and zeroth-order methods for nonconvex stochastic programming. SIAM Journal on Optimization, 23(4), 2341-2368.
>
> [14] Kingma, D. P., & Ba, J. (2015, January). Adam: A Method for Stochastic Optimization. In International Conference on Learning Representations.
>
> [15] Loshchilov, I., & Hutter, F. (2018, September). Decoupled Weight Decay Regularization. In International Conference on Learning Representations.

---

### Author Response · Authors · 2021-11-17
**General Responses**

We sincerely appreciate all reviewers for the hard work and helpful comments.

We would like to address all reviewers’ main concerns in the corresponding responses.

We also carefully revised our manuscript according to the comments.

The change we made mainly includes:
-	We presented more references and discussions, particularly about diffusion theory, Langevin Dynamics, and Fokker-Planck Equations.
-	We presented more derivation details and discussions about Adam dynamics. We presented most of them in the new Appendix H.
-	We improved some presentation details and corrected several typos.

---

### Decision · Program_Chairs · 2022-01-20

**Decision:**

Reject

**Comment:**

The paper is aimed at providing an explaining the perceived lack of generalization results for Adam as compared to SGD. To this end the paper decouples the effect of adaptive per parameter learning rate and the momentum aspect of Adam. The paper shows that the while adaptive rates help escape saddle points faster - they are worse when consider the flatness of minima being selected. Further momentum has no effect on the flatness of minima but again leads to better optimization by providing a drift leading to saddle point evasion. They also provide a new algorithm Adai (based on inertia) targeted at better generalization of adaptive methods.

The paper definitely provides an interesting perspective and the approach to decouple the effect of momentum and adaptive LR and study their efficacy in escaping saddle points and flatness of minima seems a very useful perspective. The primary reason for my recommendation is the presentation of the paper in terms of the rigor its assumptions to establish the results. These aspects have been highlighted by the reviewers in detail. I suggest the authors to carefully revisit the paper and improve the presentation of the assumptions, adding rigor to the presentation as well as adding justifications where appropriate especially in light of non-standardness of these assumptions in optimization literature.